# Altered hippocampal-prefrontal communication during anxiety-related avoidance in mice deficient for the autism-associated gene *Pogz*

Margaret M Cunniff, Eirene Markenscoff-Papadimitriou, Julia Ostrowski, John LR Rubenstein, Vikaas Singh Sohal*

Department of Psychiatry, Weill Institute for Neurosciences, and Kavli Institute for Fundamental Neuroscience, University of California, San Francisco, San Francisco, United States

**Abstract** Many genes have been linked to autism. However, it remains unclear what long-term changes in neural circuitry result from disruptions in these genes, and how these circuit changes might contribute to abnormal behaviors. To address these questions, we studied behavior and physiology in mice heterozygous for *Pogz*, a high confidence autism gene. $Pogz^{+/-}$ mice exhibit reduced anxiety-related avoidance in the elevated plus maze (EPM). Theta-frequency communication between the ventral hippocampus (vHPC) and medial prefrontal cortex (mPFC) is known to be necessary for normal avoidance in the EPM. We found deficient theta-frequency synchronization between the vHPC and mPFC in vivo. When we examined vHPC–mPFC communication at higher resolution, vHPC input onto prefrontal GABAergic interneurons was specifically disrupted, whereas input onto pyramidal neurons remained intact. These findings illustrate how the loss of a high confidence autism gene can impair long-range communication by causing inhibitory circuit dysfunction within pathways important for specific behaviors.

*For correspondence:
vikaas.sohal@ucsf.edu

## Introduction

Mutations in *Pogz* have been identified in over forty patients with autism spectrum disorder (ASD) (*Fukai et al., 2015*; *Hashimoto et al., 2016*; *Iossifov et al., 2012*; *Iossifov et al., 2014*; *Neale et al., 2012*; *Stessman et al., 2016*; *Zhao et al., 2019*), intellectual disability (*Dentici et al., 2017*; *Fitzgerald et al., 2015*; *Gilissen et al., 2014*; *Tan et al., 2016*; *White et al., 2016*; *Ye et al., 2015*), and schizophrenia (*Fromer et al., 2014*; *Gulsuner et al., 2013*). Most of these are de novo mutations presumed to cause loss of function. Such de novo loss of function mutations are exceedingly rare in controls, ranking *Pogz* among the highest confidence genes for ASD (FDR < 0.01) (*Sanders et al., 2015*). POGZ is known to play a role in chromatin regulation, mitotic progression, and chromosome segregation (*Nozawa et al., 2010*). ASD associated mutations have been shown to disrupt POGZ's DNA-binding activity (*Matsumura et al., 2016*) and reduce neurite outgrowth in vitro (*Hashimoto et al., 2016*; *Zhao et al., 2019*).

Among the highest confidence ASD-associated genes, there is a striking enrichment for genes which, like *Pogz*, are involved in chromatin remodeling (*Cotney et al., 2015*; *De Rubeis et al., 2014*; *Krumm et al., 2014*; *Sanders et al., 2015*). One hypothesis is that this enrichment reflects the developmental complexity of the nervous system, which renders the brain more vulnerable than other systems to regulatory disruptions (*Ronan et al., 2013*; *Suliman et al., 2014*). This hypothesis is supported by the convergent expression of genes associated with neurodevelopmental disease at specific developmental timepoints (*Gulsuner et al., 2013*; *Willsey et al., 2013*). Despite this

progress in identifying ASD-associated genes and their convergence onto specific developmental processes, we do not yet understand how these genetic disruptions cause behavioral phenotypes, nor what mechanisms in the developed brain might be targeted to normalize behavior. This is because it remains unclear what long-term changes in neural circuitry result from these genetic disruptions, and how they might contribute to the abnormal functioning of the developed brain.

In order to further understand the nature of neural network dysfunction that results from genetic disruptions and altered development, we characterized behavior and physiology in adult *Pogz* heterozygous loss of function (*Pogz*$^{+/-}$) mice. We found that these mice exhibit altered behavior in a well-studied assay of anxiety-related avoidance, the elevated plus maze (EPM). Interestingly, this is similar to a recently published study which found evidence for decreased anxiety in *Pogz* mutants using the open field test (*Matsumura et al., 2020*). We then studied communication between the ventral hippocampus (vHPC) and medial prefrontal cortex (mPFC), which is known to be necessary for normal anxiety-related avoidance in the EPM. We found that theta-frequency synchronization between the vHPC and mPFC is decreased in vivo. In vitro, we found a specific loss of excitatory synaptic drive from the vHPC onto prefrontal GABAergic interneurons. Notably excitatory input from the vHPC onto prefrontal pyramidal neurons was spared.

Two major hypotheses about the pathophysiology of ASD are that developmental disruptions can lead to (1) persistent dysfunction of cortical GABAergic circuits (*Nelson and Valakh, 2015*; *Rubenstein and Merzenich, 2003*; *Sohal and Rubenstein, 2019*), and (2) impairments in long-range communication (*Kana et al., 2014*). Our findings illustrate a case in which these two mechanisms may be linked following the heterozygous loss of a high confidence ASD gene – specifically, deficient long-range communication is associated with an impairment in inhibitory circuits. Our findings also suggest that feedforward inhibition (not just feedforward excitation) in the vHPC-mPFC pathway may play an important role in anxiety-related avoidance behavior.

## Results

### Pogz$^{+/-}$ mice have decreased anxiety-related avoidance in the EPM

To characterize their behavioral phenotypes, we tested *Pogz*$^{+/-}$ mice using a battery of standard behavioral assays. We found a reduction in anxiety-related avoidance in the EPM (*Figure 1A,B*). Rodents typically avoid the center and open arms of the EPM, because they are exposed, brightly lit, and raised off the ground, and instead spend the bulk of their time in the closed arms. However, *Pogz*$^{+/-}$ mice spent significantly more time exploring the open arms and center region of the EPM compared to their wildtype littermates (*Figure 1C,E*; ratio of open vs. closed arm time: p=0.003; open time: p=0.001, center time: p=0.02. Wilcoxon rank sum, WT N = 18, Het N = 27). The total distance traveled during the assay was not different between genotypes, suggesting that this increase in open arm exploration is not simply an artefact related to changes in overall exploratory behavior (*Figure 1D*, p=0.35, Wilcoxon rank sum, WT N = 16, Het N = 23). *Pogz* heterozygotes also made more head dips in the EPM than their wildtype littermates, consistent with the interpretation that their phenotype reflects a decrease in anxiety-related behavior and a corresponding increase in active exploration (*Figure 1F*, p=0.03, Wilcoxon rank sum, WT N = 14, Het N = 14). There was no difference in the number of open arm entries between genotypes, but individual open arms visits were longer in duration in *Pogz*$^{+/-}$ mice (*Figure 1G,H*; number of entries: p=0.32; duration of entries: p=0.047, Wilcoxon rank sum, WT N = 16, Het N = 23). We also confirmed that increases in open arm exploration and head dips were not driven simply by sex differences (*Figure 1—figure supplement 1*). The performance of *Pogz* heterozygotes did not differ from that of wild-type mice on cognitive tests including an odor-texture rule shifting task (*Cho et al., 2015*; *Ellwood et al., 2017*) and a T-maze based delayed nonmatch-to-sample task (*Spellman et al., 2015*; *Tamura et al., 2017*). This indicates that their altered behavior in the EPM was not related to nonspecific impairments in spatial cognition or learning (*Figure 1—figure supplement 2*).

### Pogz$^{+/-}$ mice have reduced hippocampal-prefrontal theta synchrony

Many studies, including work from our lab, have shown that communication between the vHPC and medial prefrontal cortex (mPFC), is necessary for anxiety-related avoidance in the EPM, and that theta-frequency synchronization between these structures can serve as a biomarker for this

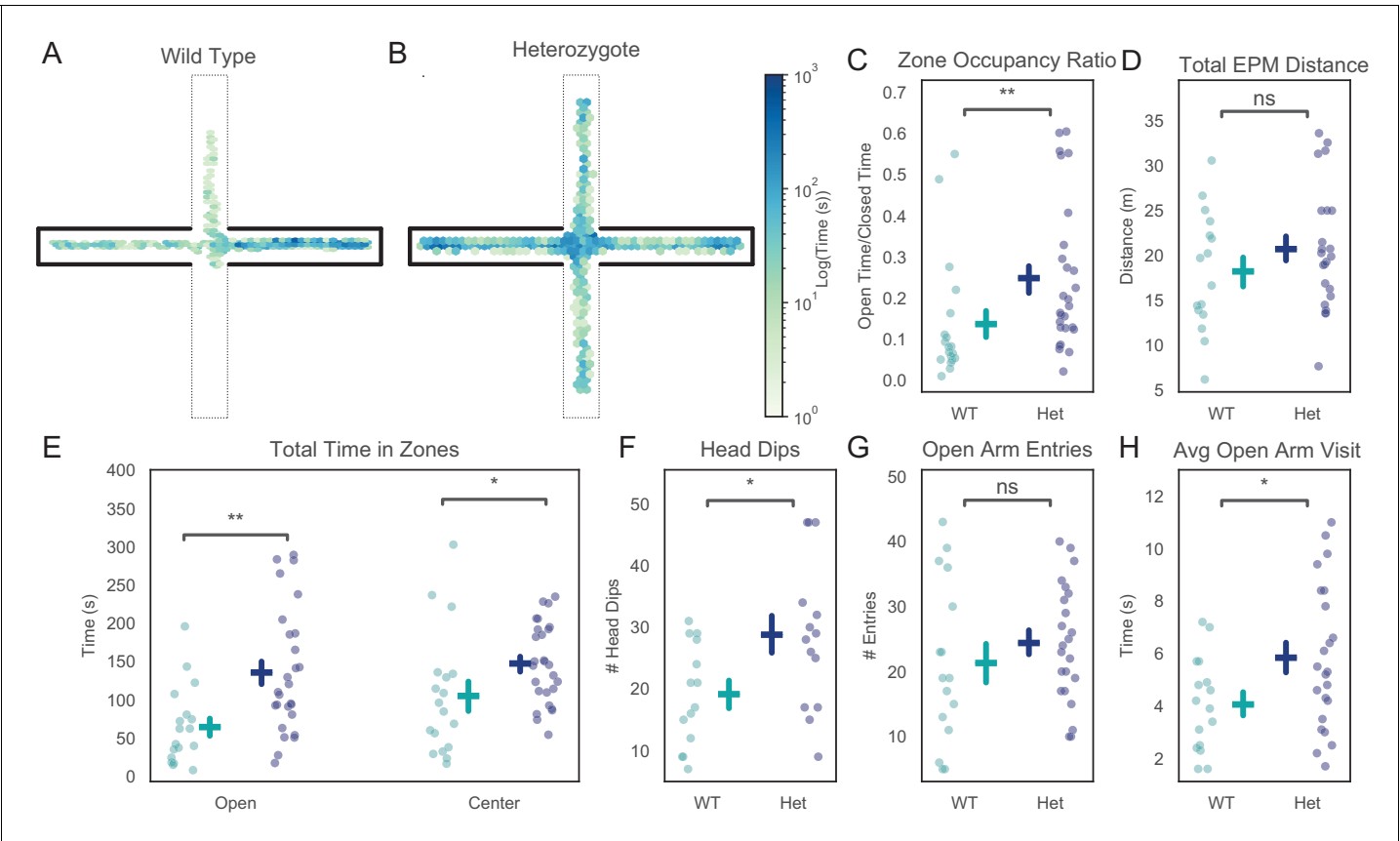

**Figure 1.** *Pogz*+/- mice exhibit reduced avoidance in the elevated plus maze (EPM). (**A, B**) Occupancy plot for a 15 min EPM session for a representative wildtype (**A**) and *Pogz*+/- (**B**) mouse. (**C**) Ratio of time spent in open vs. closed arms of the EPM. Wilcoxon rank-sum test, U = −2.8857, p=0.003, WT N = 18, Het N = 27. (**D**) Total distance traveled during EPM sessions. Wilcoxon rank-sum test U = −1.9434, p=0.35, WT N = 16, Het N = 23. (**E**) Total time spent in exposed areas of EPM, open arms: statistic = −3.0753, p=0.001, center: U = −2.2112, p=0.02. Wilcoxon rank-sum test, WT N = 18, Het N = 27. (**F**) Total number of head dips for each mouse, U = −1.9434, p=0.03. Wilcoxon rank-sum test, WT N = 14, Het N = 14. (**G**) Number of open arm entries, U = −0.9993, p=0.32. Wilcoxon rank-sum test, WT N = 16, Het N = 23. (**H**) Average duration of each open arm visit, U = −1.984, p=0.047. Wilcoxon rank-sum test, WT N = 16, Het N = 23.

The online version of this article includes the following source data and figure supplement(s) for figure 1:

**Source data 1.** Source data for *Figure 1*.
**Figure supplement 1.** Sex differences do not account for elevated-plus maze (EPM) phenotypes.
**Figure supplement 2.** Other behavioral assays in *Pogz*+/- mice.
**Figure supplement 3.** Distributions of sex and age for WT and Pogz Het mice used in all experiments.

communication (*Adhikari et al., 2010*; *Adhikari et al., 2011*; *Jacinto et al., 2016*; *Kjaerby et al., 2016*; *Lee et al., 2019*; *Padilla-Coreano et al., 2016*; *Padilla-Coreano et al., 2019*). Based on this, we recorded local field potentials from the mPFC and vHPC to assess hippocampal-prefrontal theta synchrony in *Pogz*+/- mice (*Figure 2A*). Because we were specifically interested in hippocampal-theta prefrontal synchrony, we limited analysis to mice which had clearly visible theta-frequency peaks in vHPC power spectra recorded during periods of locomotion, and had electrodes located within the mPFC and vHPC (based on post-hoc histology). Previous work has shown that vHPC-mPFC theta synchrony is dynamically modulated in different compartments of the EPM (*Adhikari et al., 2010*; *Jacinto et al., 2016*). Consistent with these earlier findings, in wild-type mice, vHPC-mPFC theta synchrony increased as mice approached the center of the EPM. This has previously been interpreted to reflect movement from a less-anxiogenic to more anxiogenic location, as well as the approach to a choice point where mice must decide whether to avoid or explore the open arms (*Adhikari et al., 2010*; *Jacinto et al., 2016*). We measured synchrony between the vHPC and mPFC using the weighted-phase locking index (WPLI) (*Vinck et al., 2011*) and found that the increase in theta synchrony, which normally occurs as mice approach the center of the EPM, was conspicuously

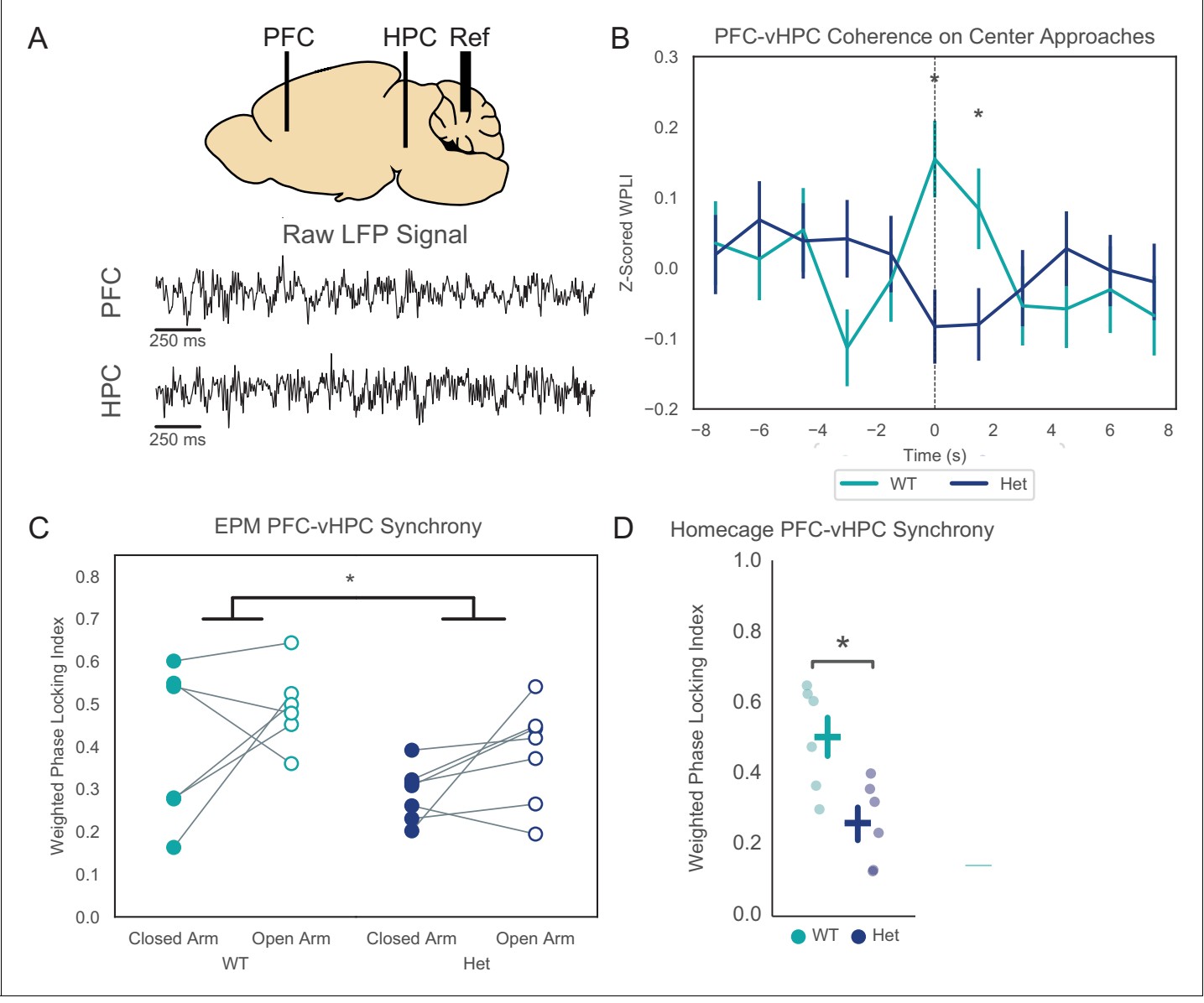

**Figure 2.** *Pogz*$^{+/-}$ mice have reduced vHPC-PFC theta synchrony both at baseline and in the elevated-plus maze (EPM). (**A**) Recording schematic and examples of raw local field potential traces. (**B**) Z-scored theta band weighted-phase locking index (WPLI) as mice approach the center of the EPM. Linear mixed effects model using timepoint (−3,–1.5, 0, and +1.5 s relative to center zone entry), genotype, mouse, and timepoint X genotype interaction as fixed factors and individual run as a random factor, p=0.00039 for timepoint X genotype interaction, t-statistic = − 3.55, DF = 2355 for fixed factors, n = 274 and 316 closed-center runs from 7 WT mice and 6 Het mice, respectively. Wilcoxon rank-sum test for t = 0: U = 3.3738, p=0.0007, for t = 1.5: U = 2.0275, p=0.043 (n = 274 closed-center runs from 7 WT mice and 316 from 6 Het mice). (**C**) Average theta band WPLI in the open vs. closed arms of the EPM. Two-way ANOVA including arm and genotype as factors - significant effect of genotype: p=0.03 (d.f. = 1, N = 6 WT and 7 Het mice, F = 5.66). (**D**) Theta band WPLI for mice in their homecages: U = 2.2417, p=0.031 (Wilcoxon rank-sum with N = 6 WT and 7 Het mice).
The online version of this article includes the following source data and figure supplement(s) for figure 2:

**Source data 1.** Source data for *Figure 2*.
**Figure supplement 1.** LFP power in various frequency bands in the vHPC and mPFC is not changed in *Pogz*$^{+/-}$ mice.
**Figure supplement 2.** Location of LFP electrodes (**A–C**) mPFC electrode locations.

absent in *Pogz* heterozygous mice, (***Figure 2B***; p=0.00039 for genotype X timepoint as a fixed factor in a linear mixed model; difference in theta synchrony at the time of center approach: p=0.001). *Pogz*$^{+/-}$ mice also had overall reduced vHPC-mPFC theta synchrony while in the EPM, as compared to wild-type littermates (***Figure 2C***; 2-way ANOVA with genotype and open vs. closed arms as

factors, significant effect of genotype, p=0.03). In fact, *Pogz*$^{+/-}$ mice had reduced theta synchrony at baseline, in the home cage (*Figure 2D*; p=0.03). There were no differences in power in the vHPC or mPFC between *Pogz*$^{+/-}$ mice and wildtypes, suggesting that this change in synchrony reflects altered communication between these brain regions, not just reduced activity in one or both structures (*Figure 2—figure supplement 1*).

Notably, WPLI is unsigned, that is, it measures phase locking using the magnitudes of the imaginary component of the phase difference. However, when we examined the signs of these phase differences, we found that when mice were in the open arms, for 5/6 wild-type mice and 6/6 *Pogz* heterozygous mice, the imaginary component of the phase difference was above the x-axis in the complex plane, indicating that hippocampal activity tends to lead prefrontal activity.

## An unbiased, data-driven approach to examine the significance of vHPC-mPFC theta synchrony for normal behavior and Pogz$^{+/-}$ mice

As noted earlier, many studies have focused on vHPC-mPFC theta synchrony as a potential biomarker for vHPC-mPFC communication that is relevant to anxiety-related behaviors. As described above, we found deficits in vHPC-mPFC theta synchrony that correlate with deficits in anxiety-related avoidance behaviors in *Pogz*$^{+/-}$ mice. However, perhaps this is simply a case of the streetlight effect. I.e., perhaps there are alternative patterns of activity within the hippocampal-prefrontal circuit that are also engaged during EPM exploration, but which remain largely intact in *Pogz*$^{+/-}$ mice. In this context, multiple studies from the Dzirasa laboratory and one from ours have shown that data-driven approaches can uncover patterns of rhythmic activity across limbic networks ('electomes' or 'intrinsic coherence networks') which correlate with, and potentially predict, aspects of emotional behaviors (*Hultman et al., 2016*; *Hultman et al., 2018*; *Kirkby et al., 2018*). Can this kind of data-driven approach identify hippocampal-prefrontal networks that are engaged by EPM exploration, and if so, would these be intact or deficient in *Pogz*$^{+/-}$ mice?

To address this question, we took a data-driven approach to identify salient features within LFP recordings, relate these to EPM behavior, and assess them in *Pogz*$^{+/-}$ mice. A combination of principal components analysis (PCA) (to compute dimensionality) and independent components analysis (ICA) (to reduce dimensionality) was applied (Methods) to a broad list of potential LFP features for all mice (*Table 1*, *Table 2*). These features comprise power (within each region), synchrony (between regions), and cross-frequency coupling (within or between regions), across multiple frequency bands. Each independent component (ICs) discovered in this way was defined by a set of weights for each feature (*Figure 3A*; 80 total ICs derived from 13 mice). To identify similar ICs that were conserved across mice and thus likely to be biologically meaningful, we calculated the correlation coefficient between all pairs of ICs (*Figure 3B*), then applied a threshold to this pairwise correlation matrix to identify pairs of highly similar ICs (*Figure 3C*). We then performed clustering on this dataset (Methods) to identify characteristic ICs that appear repeatedly across mice. One such cluster was characterized by strong weights for cross-frequency (phase-amplitude) coupling between hippocampal-theta and higher-frequency activity in either the vHPC or mPFC (*Figure 3D*). In other words, this cluster corresponds to a 'network' that is conserved across mice. When activity in this network goes up, it means that the hippocampal-theta rhythm more strongly modulates the amplitude of beta and gamma-frequency activity in both the hippocampus and prefrontal cortex.

For each mouse, we could calculate the time-varying activity of this IC by convolving the weights of this IC (averaged across mice) with the time series of each feature. When mice approach the center of the EPM during closed arm-center-open arm runs, the activity of this IC shows a pattern similar to what we previously observed for vHPC-mPFC theta synchrony. Specifically, in wild-type mice, the activity of this IC increased as mice ran approached the center zone. Strikingly, this behavioral modulation of network activity was once again absent in *Pogz* heterozygotes (*Figure 3E*). Thus, this unbiased approach validated the general finding we made earlier, when we focused on a specific metric of vHPC–mPFC theta synchrony. Again we found a pattern of activity, related to theta-frequency synchronization across the hippocampal-prefrontal circuit (in this case, measured by the modulation of higher-frequency activity), normally correlates with entries into more anxiogenic regions of the EPM, but this relationship is abolished in *Pogz* heterozygotes.

We found a total of three clusters, corresponding to three characteristic ICs that were conserved across mice. The average weights for each of these characteristic ICs, as well as the pattern of activity for each during approaches to the center of the EPM, are plotted in *Figure 3—figure supplement 1*.

**Table 1.** Single frequency LFP measures used as features in PCA/ICA analysis.

| Measure | Region | Frequencies |
|---|---|---|
| Power | HPC | Theta (4–12 Hz) |
| | | Beta (13–30 Hz) |
| | | Low Gamma (30–55 Hz) |
| | | High Gamma (65–100 Hz) |
| | PFC | Theta (4–12 Hz) |
| | | Beta (13–30 Hz) |
| | | Low Gamma (30–55 Hz) |
| | | High Gamma (65–100 Hz) |
| Amplitude Covariation | HPC-PFC | Theta (4–12 Hz) |
| | | Beta (13–30 Hz) |
| | | Low Gamma (30–55 Hz) |
| | | High Gamma (65–100 Hz) |
| Weighted-Phase Locking | HPC-PFC | Theta (4–12 Hz) |
| | | Beta (13–30 Hz) |
| | | Low Gamma (30–55 Hz) |

High Gamma (65–100 Hz).

As described above, one characteristic IC represents coupling between vHPC theta phase and the amplitude of higher-frequency vHPC or mPFC activity. Another characteristic IC represents coupling between mPFC theta phase and the amplitude of higher-frequency vHPC or mPFC activity. Notably, activity in the latter characteristic IC was not appreciably modulated during approaches to the center of the EPM. The third characteristic IC represents broadband vHPC and mPFC power. We also did not find a characteristic IC corresponding to coupling between alpha phase and higher-frequency activities. These observations support our finding that theta-frequency communication between the hippocampus and downstream structures such as the PFC is behaviorally modulated, and that the normal pattern of modulation is disrupted in *Pogz* mutant mice. Notably, this finding is specific for both frequency band and anatomical pathway, as we did not find conserved clusters of ICs corresponding to cross-frequency coupling outside the theta band and did not observe behavioral modulation for the IC which represents coupling of vHPC activity to mPFC theta.

### vHPC excitation of mPFC interneurons is deficient in Pogz$^{+/-}$ mice

Impaired synchrony suggests a deficit in the transmission of neural activity from the vHPC to mPFC. This could reflect local deficits within these structures, and/or altered synaptic connections between them. To explore potential factors underlying this impaired synchrony, we made patch clamp recordings from neurons in the prefrontal cortex. The resting membrane potential, input resistance, and action potential properties of pyramidal cells and interneurons were not grossly different between *Pogz*$^{+/-}$ mice and wild-type littermates (*Figure 4—figure supplement 1*, *Figure 5—figure supplement 1*). To assess synaptic communication between the vHPC and mPFC, we injected virus encoding CamKII-ChR2-EYFP into the vHPC, then, after waiting 8 weeks for viral expression, recorded optically-evoked responses in the mPFC. Optogenetic stimulation was delivered at 8 Hz, to specifically focus on theta-frequency transmission. We recorded both excitatory currents and optically-evoked spikes (*Figure 4*).

Fast-spiking interneurons (FSINs) in *Pogz* heterozygotes showed a marked reduction in excitatory synaptic input from vHPC projections, including a ~ 50% reduction in total charge (*Figure 4*, p=0.006, Wilcoxon rank sum WT N = 6, n = 11, Het N = 3, n = 7). Short term plasticity of these excitatory synapses onto FSINs also exhibited a shift toward greater depression as evidenced by a decrease in the paired-pulse ratio (PPR) (*Figure 4*, p=0.03, Wilcoxon rank sum WT N = 6, n = 11, Het N = 3, n = 7). In current clamp recordings, these FSINs exhibited a much longer latency to spike following each light flash (*Figure 4*, p=0.01, Wilcoxon rank sum WT N = 6, n = 11, Het N = 3, n = 7). There was a trend toward an overall reduction in spiking which did not reach statistical significance

**Table 2.** Multiple frequency LFP measures used as features in PCA/ICA analysis.

| Measure | Regions | Frequencies |
|---|---|---|
| Cross-Frequency Coupling | HPC (low) → PFC (high) | Theta (2–6 Hz) → Beta (13–30 Hz) |
| | | Theta (2–6 Hz) → Low Gamma (30–55 Hz) |
| | | Theta (2–6 Hz) → High Gamma (65–100 Hz) |
| | | Alpha (6–10 Hz) → Beta (13–30 Hz) |
| | | Alpha (6–10 Hz) → Low Gamma (30–55 Hz) |
| | | Alpha (6–10 Hz) → High Gamma (65–100 Hz) |
| | PFC (low) → HPC (high) | Theta (2–6 Hz) → Beta (13–30 Hz) |
| | | Theta (2–6 Hz) → Low Gamma (30–55 Hz) |
| | | Theta (2–6 Hz) → High Gamma (65–100 Hz) |
| | | Alpha (6–10 Hz) → Beta (13–30 Hz) |
| | | Alpha (6–10 Hz) → Low Gamma (30–55 Hz) |
| | | Alpha (6–10 Hz) → High Gamma (65–100 Hz) |
| | HPC (low) → HPC (high) | Theta (2–6 Hz) → Beta (13–30 Hz) |
| | | Theta (2–6 Hz) → Low Gamma (30–55 Hz) |
| | | Theta (2–6 Hz) → High Gamma (65–100 Hz) |
| | | Alpha (6–10 Hz) → Beta (13–30 Hz) |
| | | Alpha (6–10 Hz) → Low Gamma (30–55 Hz) |
| | | Alpha (6–10 Hz) → High Gamma (65–100 Hz) |
| | PFC (low) → PFC (high) | Theta (2–6 Hz) → Beta (13–30 Hz) |
| | | Theta (2–6 Hz) → Low Gamma (30–55 Hz) |
| | | Theta (2–6 Hz) → High Gamma (65–100 Hz) |
| | | Alpha (6–10 Hz) → Beta (13–30 Hz) |
| | | Alpha (6–10 Hz) → Low Gamma (30–55 Hz) |

Alpha (6–10 Hz) → High Gamma (65–100 Hz).

(*Figure 4*, p=0.08, Wilcoxon rank sum WT N = 6, n = 11, Het N = 3, n = 7). Notably, all of these changes were specific to FSINs. In recordings from pyramidal neurons, we did not observe any changes in the size or PPR of optogenetically evoked synaptic currents, nor in the latency or number of optogenetically evoked spikes (*Figure 5*).

## Deficient FSIN excitation impairs information transmission across vHPC-mPFC circuits

Excitatory and inhibitory postsynaptic currents are major contributors to LFPs (*Buzsáki et al., 2012*). Thus, a major deficit in synaptic currents evoked by hippocampal inputs could explain the reductions

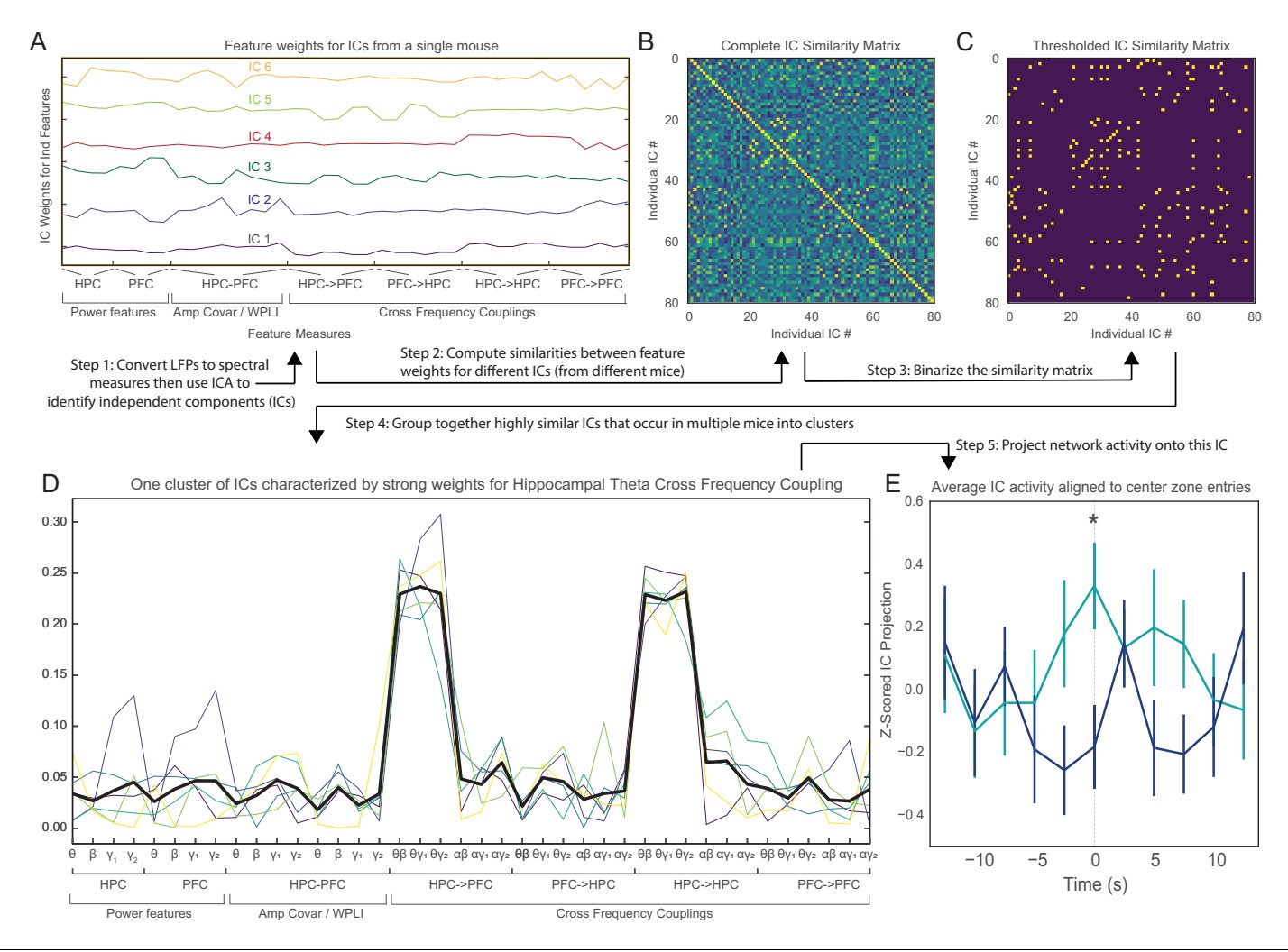

**Figure 3.** An unbiased, data-driven approach confirms that theta-frequency vHPC-mPFC communication is behaviorally-relevant and deficient in *Pogz*[+/-] mice. (**A**) Example weight vectors showing how various LFP features (x-axis) contribute to different independent components (ICs) in one mouse. The y-axis shows the weight of each feature. (**B**) Correlation matrix showing the similarity of weight vectors corresponding to different ICs, from all mice. (**C**) Binarized version of the correlation matrix showing pairs of ICs that have a correlation coefficient > 0.7. (**D**) Example weights vectors (light, colored traces) for ICs from one cluster. This cluster is characterized by strong weights for cross-frequency coupling between vHPC theta activity and higher-frequency activity in either vHPC or mPFC. The bold black trace shows the average of these weight vector. (**F**) The projection of network activity onto the characteristic (averaged) weight vector (from E) as a function of time during approaches to the center of the EPM, for wild-type or *Pogz*[+/-] mice. As mice approach the center, activity in this characteristic IC rises sharply and reaches a peak in WT mice, but this is absent in *Pogz*[+/-] mice. Linear mixed effects model using timepoints (t = 0 vs. baseline based on the average of the first/last points), mouse, genotype, and timepoint X genotype interaction as fixed factors, and individual runs as random factors, timepoint X genotype interaction p=0.01, DF = 147, t-statistic = 2.60; Wilcoxon rank-sum test for t = 0: p=0.007, U = 2.6864; n = 39 closed-center-open runs from 6 WT mice and 37 runs from 7 Het mice.

The online version of this article includes the following figure supplement(s) for figure 3:

**Figure supplement 1.** Activity in conserved independent components (ICs) during approaches to the center of the EPM.

in synchronization between vHPC and mPFC LFPs that we observed. But how might this synaptic deficit in *Pogz*[+/-] mice explain their decreased avoidance of the open arms in the EPM? As discussed above, the transmission of information from the vHPC to mPFC is necessary for open arm avoidance. We hypothesized that a decrease in excitatory drive onto FSINs could impair the PFC's ability to appropriately filter information, reducing the transmission of information from the vHPC to mPFC, and resulting in the decreased open arm avoidance seen in *Pogz* heterozygotes. Specifically, we hypothesized that because ventral hippocampal input to the mPFC is rhythmically modulated,

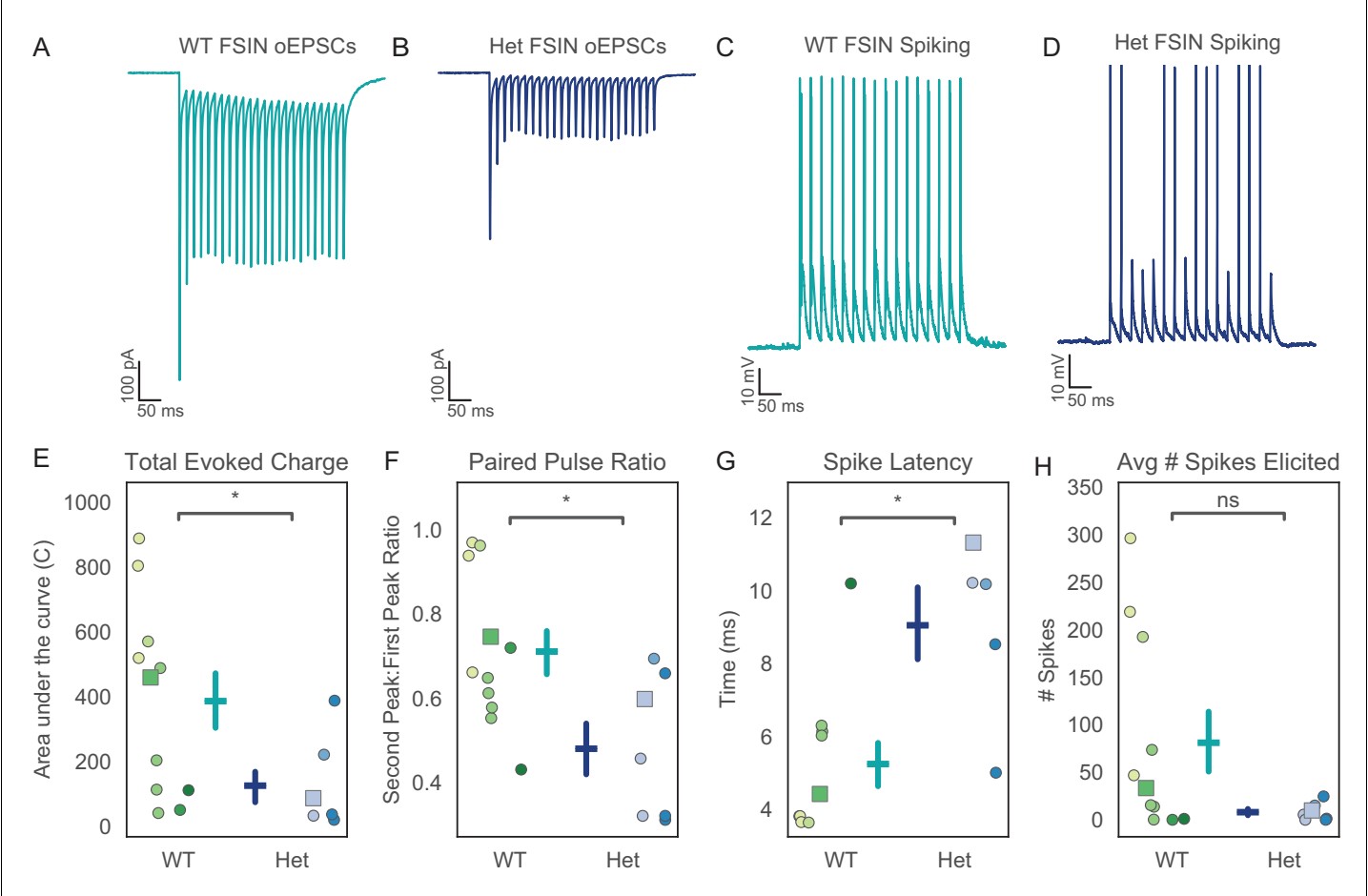

**Figure 4.** Excitatory hippocampal input to prefrontal fast-spiking interneurons (FSINs) is reduced in *Pogz* mutants. (**A, B**) Representative examples of optically-evoked excitatory post-synaptic currents (oEPSCs) recorded from prefrontal FSINs in wildtype (**A**) or *Pogz*[+/-] mice (**B**). (**C, D**) Representative traces of optically-evoked excitatory post-synaptic potentials (oEPSPs) and action potentials recorded from FSINs in wildtype (**C**) or *Pogz*[+/-] mice (**D**). (**E**) The total oEPSC charge in FSINs is reduced in *Pogz*[+/-] mice, U = 2.7652, p=0.006. (**F**) The paired- pulse ratio (PPR) for oEPSCs is reduced in *Pogz*[+/-] FSINs, U = 2.128, p=0.03. (**G**) The latency of the first optically-evoked action potential is increased in *Pogz*[+/-] FSINs, U = −2.490, p=0.013. (**H**) The number of action potentials elicited by oEPSPs is non-significantly altered, U = 1.766, p=0.08. In **E-H**, different hues correspond to specific mice, and squares indicate datapoints from cells that were used for the representative traces shown in **A-D**. All p-values from Wilcoxon rank sum, WT N = 6 animals, n = 11 cells. Het N = 3 animals, n = 7 cells.

The online version of this article includes the following source data and figure supplement(s) for figure 4:

**Source data 1.** Source data for *Figure 4*.

**Figure supplement 1.** Intrinsic properties of prefrontal FSIN are not changed in *Pogz*[+/-] mice.

feedforward inhibition might preferentially suppress the responses of prefrontal neurons to out-of-phase 'noise' while sparing hippocampally-driven responses.

To test the plausibility of this hypothesis, we constructed a simple computational model composed of 2 integrate-and-fire neurons – a FSIN and an output neuron (i.e. a pyramidal cell). Both cells received the same two sources of synaptic input – 'noise,' generated by a Poisson process with constant rate, and 'hippocampal input,' which was modeled as a Poisson process whose rate varied according to the theta rhythm, that is, was modulated at 8 Hz (*Figure 6A*). Both cells had the same thresholds and membrane time constants, and we set the time constants of decay for EPSPs and IPSPs to 8 and 20 msec, respectively, to reflect the typically longer timescales for synaptic inhibition. The rate of hippocampal inputs varied sinusoidally between 0 and 100 Hz, and the rate of noise inputs was constant at the midpoint of this distribution (50 Hz). Pyramidal neuron spiking ranged from ~0 to 50 Hz, whereas FSIN spiking ranged from ~0 to 150 Hz. Finally, we explored how varying the strength of excitatory input from both hippocampal and noise inputs onto FSINs affected the

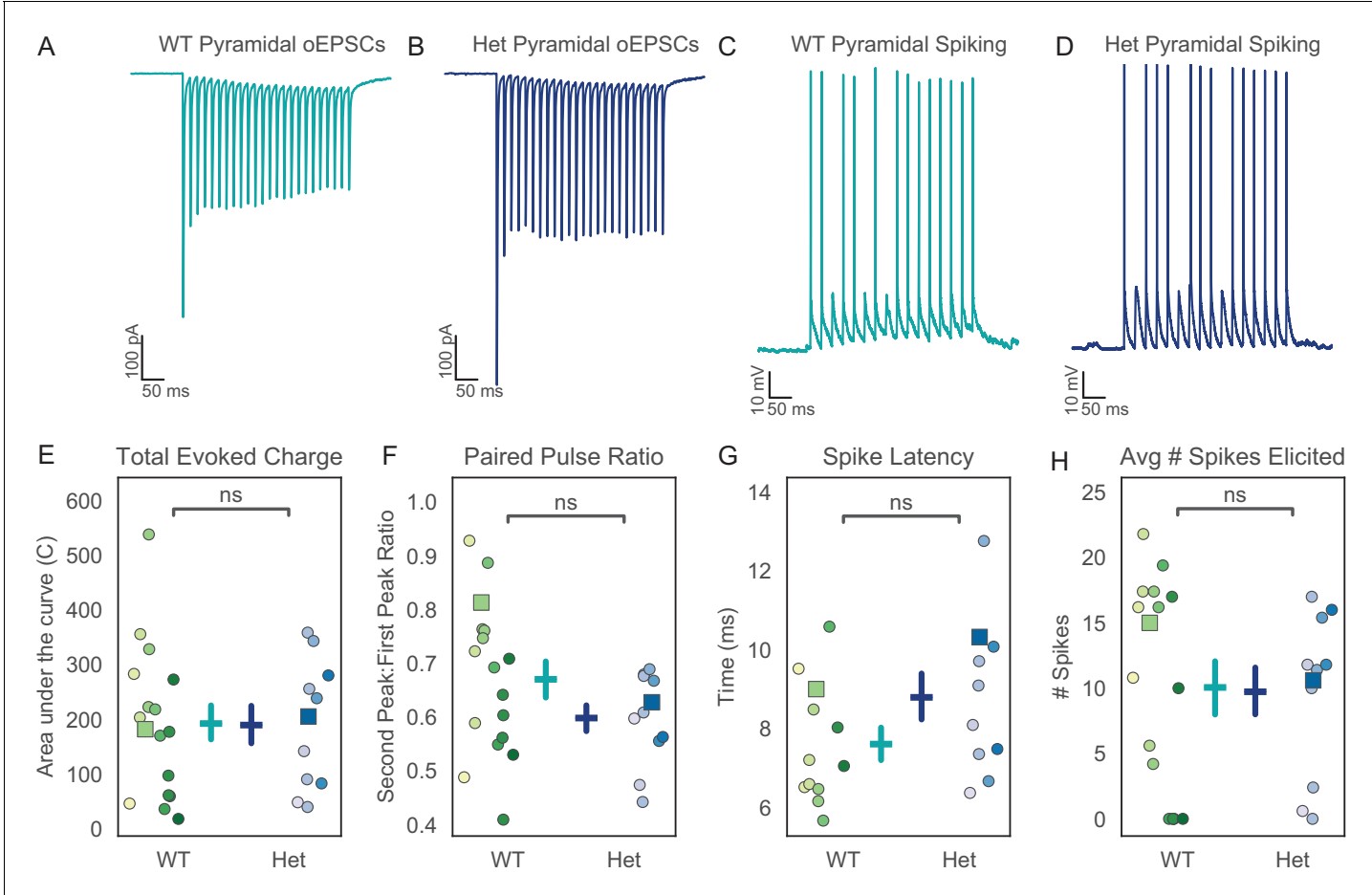

**Figure 5.** Excitatory hippocampal input to prefrontal pyramidal neurons is not changed in *Pogz* mutants. (A, B) Representative examples of optically-evoked excitatory post-synaptic currents (oEPSCs) recorded from prefrontal pyramidal neurons in wildtype (A) or *Pogz*$^{+/-}$ mice (B). (C, D) Optically-evoked excitatory post-synaptic potentials (oEPSPs) and action potentials in wildtype (C) or *Pogz*$^{+/-}$ (D) pyramidal neurons. (E) Total oEPSC charge in pyramidal neurons, U = 1.0736, p=0.28. (F) Paired-pulse ratio for oEPSCs in pyramidal neurons, U = 1.4347, p=0.15 (G) Latency to first optically-evoked action potential in pyramidal neurons, U = −0.305, p=0.76. (H) Number of action potentials elicited by oEPSPs in pyramidal neurons, U = 0.2822, p=0.78. In E-H, different hues correspond to specific mice, and squares indicate datapoints from cells that were used for the representative traces shown in A-D. All p-values from Wilcoxon rank sum, WT N = 13 animals, n = 17 cells. Het N = 8 animals, n = 11 cells.

The online version of this article includes the following source data and figure supplement(s) for figure 5:

**Source data 1.** Source data for *Figure 5*.

**Figure supplement 1.** Pyramidal cell properties are not changed in *Pogz*$^{+/-}$ mice.

transmission of information from the vHPC to mPFC. Specifically, we quantified the correlation between hippocampal input and mPFC output spikes, as well as between the noise input and mPFC output spikes, while varying a single parameter which represents the EPSP amplitude that each hippocampal or noise spike elicits in the FSIN.

As expected, as excitatory drive to the FSIN decreases, the rate of FSIN spiking falls while that of the pyramidal cell goes up (*Figure 6C*). When we examined the correlation between pyramidal cell spikes and either noise or hippocampal input, we found that decreasing FSIN excitatory drive also decreases the correlation between pyramidal cell output and hippocampal input (*Figure 6B*), causing a drop in the signal-to-noise ratio (*Figure 6D*). This occurs because as the strength of FSIN excitation increases, feedforward inhibition preferentially filters noise inputs, while hippocampal inputs are spared (due to their rhythmicity) (*Figure 6B*). Thus, when FSIN excitation is weak, there is minimal FSIN spiking and minimal pyramidal cell inhibition. Under these conditions, weak input is sufficient to excite the pyramidal cell, and the circuit fails to distinguish between the rhythmically occurring hippocampal signal and the (nonrhythmic) noise. As the level of FSIN excitation increases,

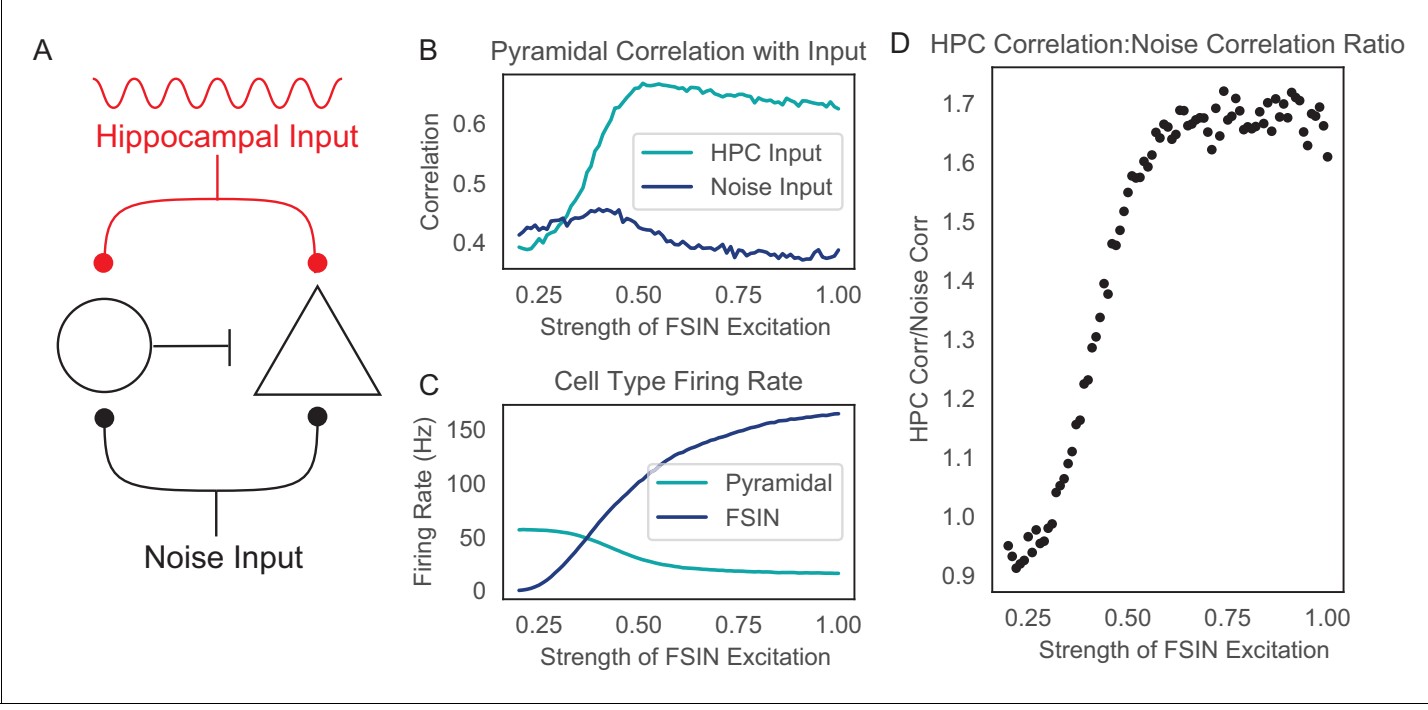

**Figure 6.** Reducing the excitatory drive onto prefrontal FSINs impairs the transmission of hippocampal inputs. (**A**) Computational model schematic. Both a model pyramidal neuron (triangle) and a model FSIN (circle) receive simulated hippocampal input (which is rhythmically modulated at 8 Hz), and additional input which represents noise. (**B**) The correlation between the pyramidal neuron output spike rate and the rate of either noise inputs (dark blue) or hippocampal spikes (turquoise), as functions of a single parameter which represents how strongly hippocampal and noise inputs excite the model FSIN. (**C**) The spike rate of the model pyramidal neuron (turquoise) and FSIN (dark blue) as functions of a single parameter representing how strongly hippocampal and noise inputs excite the model FSIN. (**D**) The ratio of the correlation between pyramidal neuron output spikes and either hippocampal input or noise input.

The online version of this article includes the following figure supplement(s) for figure 6:

**Figure supplement 1.** Adding feedforward disinhibition does not change the relationship between inhibitory strength and hippocampal correlation.

**Figure supplement 2.** The effect of reducing inhibition on the transmission of signals across hippocampal-prefrontal synapses depends on the frequency of hippocampal input.

it reaches an optimal level at which FSINs generate inhibition that suffices to filter out weak inputs. As a result, isolated noise inputs fail to elicit pyramidal cell spikes, whereas rhythmic bursts of hippocampal input provide a strong drive that allows them to be reliably transmitted via pyramidal cell spiking. Finally we note that while an extensive exploration of all possible inhibitory-disinhibitory circuit motifs is beyond the scope of this study, adding a simple form of disinhibition, in which a simulated interneuron-selective interneuron receives feedforward excitation and inhibits other interneurons, does not change our basic finding that there is an optimal level of feedforward excitation onto interneurons, below which the transmission of hippocampal input is degraded (*Figure 6—figure supplement 1*). The strength of this enhancement of hippocampal input over noise is dependent on hippocampal input frequency and best for intermediate, theta range values (*Figure 6—figure supplement 2*).

## Discussion

We identified a specific behavioral deficit in mice with heterozygous loss of function of a high confidence ASD gene, then found associated deficits in biomarkers and pathways that we and others have previous linked to this behavior. *Pogz*$^{+/-}$ mice show reduced anxiety-related avoidance in the EPM. Communication between the vHPC and mPFC is known to be necessary for this avoidance (*Kjaerby et al., 2016*; *Padilla-Coreano et al., 2016*), theta synchrony between LFPs recorded from the vHPC and mPFC is a biomarker for this communication (*Padilla-Coreano et al., 2016*), and vHPC-mPFC theta synchrony normally increases when mice approach the center of the EPM (*Adhikari et al.,*

*2010*; *Lee et al., 2019*). In *Pogz*$^{+/-}$ mice, both baseline vHPC-mPFC theta synchrony and its task-dependent modulation in the EPM are reduced. Notably, we confirmed this specific deficit in behaviorally-modulated theta-frequency vHPC–mPFC communication using an unbiased, data-driven approach. Furthermore, by directly examining vHPC-mPFC connections in brain slices, we found reduced excitatory drive from vHPC onto FSINs. This synaptic abnormality could plausibly contribute to the abnormalities we found in both avoidance behavior and LFP synchrony. Specifically, synaptic potentials and inhibitory activity are major drivers of LFP signals (*Buzsáki et al., 2012*; *Haider et al., 2016*; *Teleńczuk et al., 2017*). Thus, the deficit in vHPC excitation of mPFC interneurons we found should reduce the component of mPFC inhibitory synaptic activity that is driven by, and synchronized with, vHPC. Furthermore, we found that in a computational model, weakening feedforward excitation of inhibitory interneurons impairs the transmission of signals from the vHPC to mPFC.

Notably, a previous study found that during a working memory task, inhibiting mPFC PV interneurons did not affect vHPC-mPFC theta synchrony (*Abbas et al., 2018*). However, a recent study has found that distinct populations of vHPC pyramidal neurons project to different classes of mPFC neurons (*Sánchez-Bellot and MacAskill, 2019*). This study found that the population which specifically innervates PV interneurons also drives open arm avoidance in the EPM, whereas a distinct population of vHPC-mPFC projection neurons drives exploratory behavior. This shows that different populations of vHPC-mPFC projection neurons, which innervate distinct mPFC targets, are active during different behaviors. Thus, mPFC PV interneurons might contribute to vHPC theta synchrony during EPM behavior but not during working memory. The deficits we found in vHPC inputs to PV interneurons may also affect other interneuron populations which contribute to theta synchrony.

Interestingly, another study characterizing mice with heterozygous disruptions of *Pogz* was recently published (*Matsumura et al., 2020*). These *Pogz* mutant mice spent more time in the center of an open field, less time sniffing novel mice, and more time grooming, compared to wild-type mice. These *Pogz* mutants also exhibited an increased frequency of miniature excitatory post-synaptic currents (mEPSCs) in anterior cingulate cortex neurons. Based on the latter observation, the authors hypothesized that these *Pogz* mutants exhibit a shift in the balance of excitation and inhibition (E-I balance) toward excitation, and found that systemic treatment with an AMPA receptor antagonist increases social interaction in *Pogz* mutants. Their finding of increased time spent in the center of an open field is similar in nature to our finding that *Pogz* mutants spent increased time in the open arms of the EPM. Furthermore, we too find evidence of an alteration in E-I balance, although as elaborated below, we find that this reflects deficits in specific excitatory synapses onto inhibitory interneurons. Unlike Matsumara et al., we did not find a social deficit. However, the social assays used in the two studies were very different. Specifically, Matsumara et al. measured interaction over 60 min between the subject mouse and a novel adult mouse in an open field, whereas we measured interaction in the home cage with a junvenile mouse over 5 min. Furthermore, it is worth noting that we studied a mouse in which one copy of *Pogz* has a premature stop codon, whereas Matsumara et al. studied mice heterozygous for a patient-derived mutant allele of *Pogz*.

## vHPC–mPFC communication and anxiety

A growing body of work shows that vHPC-prefrontal communication is important for anxiety-related behavior. The vHPC, unlike other portions of the hippocampus, projects directly to prefrontal cortex (*Parent et al., 2010*), and both structures are necessary for normal anxiety-related behavior (*Kjelstrup et al., 2002*; *Shah and Treit, 2003*). Theta-frequency synchronization between activity in the vHPC and mPFC normally increases in anxiety-provoking environments such as the EPM (*Adhikari et al., 2010*). Furthermore, single units in the mPFC that encode anxiety-related information phase-lock to the hippocampal-theta rhythm more strongly than other mPFC units (*Adhikari et al., 2011*). This suggests that these anxiety-encoding prefrontal units preferentially receive theta-modulated hippocampal input. Optogenetically manipulating vHPC–mPFC projections can also bidirectionally modulate anxiety-related avoidance (*Padilla-Coreano et al., 2016*; *Padilla-Coreano et al., 2019*). In particular, suppressing vHPC input to the mPFC reduces vHPC–mPFC theta synchrony, avoidance behavior, and the encoding of anxiety-related information by mPFC neurons. In previous work, we similarly found that pharmacologically suppressing vHPC–mPFC connections reduces open arm avoidance in the EPM (*Kjaerby et al., 2016*). Our present results build on these prior findings, while also extending them in a new direction.

In particular, because inhibiting all projections from the vHPC to mPFC reduced the firing rate in the preferred arm type for mPFC neurons which prefer either the open or closed arm, a previous study concluded that the predominant effects of vHPC input to mPFC are excitatory and serve to increase firing rates in each mPFC neuron's preferred arm (*Padilla-Coreano et al., 2016*). By contrast, in *Pogz* mutant mice, the theta coordination of vHPC–mPFC activity and open arm avoidance are both impaired even though vHPC input to mPFC pyramidal neurons remains intact. This raises the possibility that feedforward inhibition may be important for vHPC input to transmit anxiety-related information to the mPFC, and that deficits in feedforward inhibition may contribute to abnormal avoidance behavior in *Pogz* mutant mice. By showing how vHPC input to interneurons and feedforward inhibition may also play an important role, our results contrast with/add to the model suggested by previous studies, in which main role of vHPC input is to provide excitation that drives mPFC neuron firing in specific locations. It is not currently possible to selectively inhibit input from one presynaptic source onto one postsynaptic cell-type (e.g., vHPC input to interneurons) using optogenetic or chemogenetic manipulations. Therefore, while imperfect, genetic models, for example, $Pogz^{+/-}$ mice, can reveal behavioral phenotypes that may result from physiological alterations that cannot be readily modeled using optogenetics or chemogenetics.

Hippocampal-prefrontal communication is important for other behaviors, besides open arm avoidance, most notably tasks that measure spatial working memory using the T-maze (*Sigurdsson et al., 2010*; *Spellman et al., 2015*). We did not find deficits in delayed alternation in $Pogz^{+/-}$ mice. This may reflect the fact that our task used a very short delay (4 s) for which prefrontal circuits may not be necessary (*Bolkan et al., 2017*), because other forms of synchronization may compensate for deficits in vHPC–mPFC theta synchrony (*Tamura et al., 2017*), or because the deficits in feedforward inhibition that we found in $Pogz^{+/-}$ mice might involve classes of prefrontal interneurons that are not required for spatial working memory (*Abbas et al., 2018*).

The other recently published study which examined mice heterozygous for a missense mutation in *Pogz* found they had smaller brains. We did not observe smaller brains, but if there were anatomical differences between our $Pogz^{+/-}$ mice and WT mice, these could have caused mistargeting of the vHPC in mutants, thereby contributing to the abnormalities we observed when measuring vHPC–mPFC synchrony. We do not believe this was the case, because we verified electrode placement both histologically (by visually examining the anatomical location of the electrode track) and electrophysiologically (by confirming the presence of a prominent theta- frequency peak in the vHPC LFP power spectrum). Importantly, the fraction of experiments excluded due to the absence of a clear theta-frequency peak in hippocampal recordings, was not different between WT and mutant mice. This suggests there was not systemic mistargeting in *Pogz* mutant mice as a result of anatomical differences.

Whereas previous studies (including our own), have taken a hypothesis-driven approach to evaluating the role of theta-frequency vHPC-mPFC communication in approach-avoidance decisions, here we also explored a data-driven approach, using ICA to identify biomarkers associated with these decisions. This approach yielded an IC which measures synchrony between theta-frequency vHPC activity and mPFC activity, and which exhibits modulation as mice approach decision points (the center zone). Thus, this IC represents a data-driven metric that shows how theta-frequency communication between the vHPC and mPFC (phase-amplitude coupling between mPFC gamma and vHPC theta) correlates with approach-avoidance decisions. Finding that this metric, like theta-frequency WPLI, is altered in $Pogz^{+/-}$ mice during closed-center transitions, thus provides strong confirmation that theta-frequency hippocampal-prefrontal communication related to approach-avoidance decisions is disrupted in $Pogz^{+/-}$ mice.

However, it is interesting to note theta phase synchrony was not itself 'pulled out' by the ICA. This presumably reflects the fact that over the entirety of the task, theta phase synchrony is being influenced by different factors than this IC, even though theta phase synchrony and this IC both evolve in parallel specifically during closed arm-center zone approaches. In other words, during closed-center runs, both theta phase synchrony and the IC both exhibit a sharp rise followed by a return to baseline. However, during the rest of the task, there must be other behaviors that differentially recruit these two measures. Future studies might identify these behaviors using approaches such as MoSeq and DeepLabCut (*Mathis et al., 2018*; *Wiltschko et al., 2015*).

A final note is that while we have measured vHPC-mPFC synchronization at the level of field potentials, an important future direction is measuring the synchronization of specific cell types, which could be done using electrophysiology or genetically encoded voltage indicators (*Cho et al., 2020*).

## Excitatory-inhibitory (E-I) balance in anxiety and autism

Another recently published study from our laboratory showed that inhibiting vasoactive intestinal polypeptide (VIP)-expressing interneurons in the mPFC causes a similar behavioral phenotype, that is, reduced open arm avoidance in the EPM (*Lee et al., 2019*). That study found VIP interneurons normally facilitate the transmission of anxiety-related information from the vHPC to mPFC by disinhibiting prefrontal responses to vHPC input. As a result, when VIP interneurons are inhibited, information about anxiety is not transmitted properly, causing mice to spend more time exploring the open arms. Since VIP interneurons inhibit other GABAergic interneurons, the effect of inhibiting VIP interneurons is to *increase* feedforward inhibition. In this context, it may seem paradoxical that the present study finds a similar phenotype (increased open arm exploration) in *Pogz*$^{+/-}$ mice when mPFC inhibition evoked by vHPC input is *impaired*. Together, these two studies underscore the importance of properly *balanced* cortical circuit inhibition.

In the context of approach-avoidance behaviors, the PFC is believed to play a key role by *evaluating* information from multiple sources in order to make a decision about whether to approach or avoid a potentially anxiogenic region (*Calhoon and Tye, 2015*). As illustrated by the computational model depicted in *Figure 6*, circuit inhibition is critical for this process. When levels of inhibition are too low, the firing of simulated mPFC output neurons is driven mainly by noise, that is, inputs unrelated to anxiety signals. This could prevent the mPFC from properly representing anxiety-related information, and/or cause the inappropriate transmission of signals related to exploratory behavior. Higher levels of inhibition can filter out the noise, allowing hippocampal inputs to be preferentially transmitted. As described in our earlier study, the ability of rhythmic hippocampal inputs to periodically recruit VIP interneuron-mediated phasic disinhibition could further promote the preferential transmission of hippocampally-driven activity. Thus, multiple classes of interneurons may work together to inhibit and filter out non-hippocampal inputs while optimizing the responsiveness to hippocampal input, potentially facilitating the transmission of anxiety-related information across hippocampal-prefrontal circuits. In this way, appropriately balanced inhibition may be indispensable for proper action selection related to approach and avoidance behaviors.

Disruptions in the balance between cortical excitation and inhibition (E-I balance) have long been hypothesized to play a role in ASD (*Lee et al., 2017*; *Rubenstein and Merzenich, 2003*). Numerous studies have identified examples of altered E-I balance related to autism. These reflect changes in the relative levels of synaptic excitation and inhibition and can be secondary to a variety of different factors, including alterations in synaptic plasticity, homeostasis, and regulatory feedback loops (*Bourgeron, 2015*; *Mullins et al., 2016*; *Nelson and Valakh, 2015*; *Sohal and Rubenstein, 2019*; *Toro et al., 2010*; *Wondolowski and Dickman, 2013*).

## Deficits in long-range communication in autism

In addition to the hypothesis that E-I balance is disturbed in autism, another hypothesis is that autism (and altered E-I balance) may reflect changes in long-range connectivity (*Just et al., 2012*). While early work focused mainly on a theory of under-connectivity in autism (*Just et al., 2004*), evidence for both hypo- and hyper-connectivity has been identified using a range of methods, including functional magnetic resonance imaging (fMRI) (*Müller et al., 2011*; *Redcay et al., 2013*), electroencephalography (EEG) (*Coben et al., 2014*; *Zeng et al., 2017*), magnetoencephalography (MEG) (*Buard et al., 2013*), and structural imaging (*Mueller et al., 2013*; *Nair et al., 2013*). Changes in long-range connectivity have been identified in a number of other disorders, including schizophrenia (*Guo et al., 2015*; *Wang et al., 2014*), generalized anxiety disorder (*Andreescu et al., 2014*; *Xing et al., 2017*), and bipolar disorder (*Kam et al., 2013*; *Wang et al., 2017*), suggesting that altered connectivity may be common to a range of neurodevelopmental and psychiatric disorders. Here we find disturbed long-range connectivity (as measured by LFP synchrony) which, when examined at a finer scale, is associated with a selective deficit in the recruitment of inhibitory interneurons. This reveals a specific mechanism – impaired feedforward inhibition – that could potentially link together two prominent hypotheses about the neurobiology of autism in a way that could contribute to behavioral abnormalities.

It should be noted that the changes we observed are not necessarily static. Connectivity abnormalities in ASD have been shown to be age-(*Keehn et al., 2013*; *Padmanabhan et al., 2013*) and state-dependent (*You et al., 2013*). Our study focuses on the outcome of developmental disruptions in the

adult brain but does not establish a direct mechanism tracing changes in *Pogz* expression to network-level changes. It is possible that these changes in connectivity would be different in juvenile mice, and/or that the changes we see reflect a compensatory response to changes at an earlier timepoint.

### Possible relevance of *Pogz* behavioral phenotypes to autism

This study focuses on a phenotype whereby *Pogz*^+/- mice exhibit reduced avoidance of the open arms of the EPM. The EPM is often regarded as an assay that measures anxiety-related behavior. In this framework, reduced open arm avoidance is interpreted to reflect reduced anxiety. Reduced anxiety is not typically associated with autism, raising a question about the relevance of our findings for the clinical condition.

On the one hand, relying on face validity to determine which mouse behavioral phenotypes are relevant to human autism can be problematic for multiple reasons. First, mouse assays measure only the most rudimentary aspects of social behavior – typically social preference and/or preference for social novelty. In many individuals with autism, social preference and preference for social novelty are intact, but social functioning is disrupted in other ways. In particular, the largest study of individuals with disruptions in *Pogz* found 'in many cases, a seemingly contrary overly social and overly friendly demeanor' (*Stessman et al., 2016*). Thus, it is questionable how well face valid mouse assays of social behavior capture the more nuanced and heterogeneous phenotypes characteristic of clinical autism. On the other hand, we do not want to assert that any behavioral phenotype observed in mice with disruptions in an autism-associated gene will automatically be relevant to autism.

In this context, a logical approach is to focus on brain regions and networks that have consistently been implicated in autism. While specific behaviors may not be well conserved across species, we hypothesize that general principles underlying the function of limbic circuits, for example, hippocampal-prefrontal interactions, will be more likely to translate. In this context, we found that prefrontal circuits fail to use limbic input to appropriately guide decisions about approach vs. avoidance behavior. This is notable as a recent review hypothesized that deficits in the ability of the prefrontal cortex to appropriate guide approach/avoidance decisions plays a key role in autism (*Pfaff and Barbas, 2019*).

### Conclusion

We characterized behavior and network-level physiology in mice with heterozygous loss of function in *Pogz*, a high confidence autism gene. *Pogz*^+/- mice show reduced avoidance behavior in the EPM and altered vHPC-PFC synchrony, consistent with recent work characterizing the role of the vHPC-mPFC circuit in anxiety behavior. Additionally, in slice experiments, we found reduced excitatory drive from the hippocampus to prefrontal FSINs, suggesting an impairment in ability to properly filter incoming hippocampal input. This work elucidates the nature of a network-level phenotype linking genetic and developmental perturbations with specific behavioral and physiological changes in the adult brain.

## Materials and methods

**Key resources table**

| Reagent type (species) or resource | Designation | Source or reference | Identifiers | Additional information |
|---|---|---|---|---|
| Strain, strain background (*Mus. Musculus*) | C57BL6/J | Jackson Labs | Stock No: 000664 | |
| Genetic reagent (*Mus. Musculus*) | *PogZ+/-* | Rubenstein Lab | | |
| Recombinant DNA reagent | AAV5-CaMKIIa-hChR2(H134R)-EYFP | UNC Vector Core | RRID:Addgene_26969 | |
| Recombinant DNA reagent | AAV5-DlxI12b-mCherry | Virovek, Sohal lab | | |

*Continued on next page*

*Continued*

| Reagent type (species) or resource | Designation | Source or reference | Identifiers | Additional information |
|---|---|---|---|---|
| Software, algorithm | Sirenia Acquisition | Pinnacle | RRID:SCR_016183 | |
| Software, algorithm | ANY-maze tracking software | ANY-maze | RRID:SCR_014289 | |
| Software, algorithm | Python | Python | RRID:SCR_008394 | Packages: Numpy, Scipy, Matplotlib, Seaborn |
| Software, algorithm | MATLAB | Mathworks | RRID:SCR_001622 | Signal Processing Toolbox |
| Software, algorithm | PClamp | Molecular Devices | RRID:SCR_011323 | |

## Subjects and behavioral assays

All experiments were conducted in accordance with procedures established by the Administrative Panels on Laboratory Animal Care at the University of California, San Francisco. Male and female mice > 4 weeks old were used in all experiments. All mice were *Pogz* heterozygotes or wild-type littermates. Gene expression changes in these mice are characterized in a related publication (Markenscoff-Papadimitriou et al., in preparation). Briefly, these mice were generated by CRISPR-Cas9 and sgRNAs targeting exons 1 and 6, a 10 kb span, which generated a premature stop codon. Reduced *POGZ* expression in *Pogz*$^{+/-}$ cortex at P28 was verified by Western blot.

Unless otherwise noted, experiments were performed under ambient light and mice were group housed with littermates. Mice were habituated to the behavioral testing area for >30 min at the beginning of all sessions. LFP recording during behavior was done in a separate cohort from the mice used to establish behavioral phenotypes. For LFP experiments, mice were habituated to the head tether in their home cage for 15 min daily for 3 days. ANY-maze (Stoelting) was used to track the position of the mouse during assays using a USB webcam. Experimenter was blinded to each mouse's genotype during behavioral assessment. Note: The overall design for our behavioral studies was to perform an initial screen using multiple behavioral assays. This initial screen revealed altered behavior in the EPM, but not for many other behavioral assays, for example, for social interaction. Therefore, we then we validated the EPM finding using additional mice. For this reason, the *N* is larger for the EPM than for other social and cognitive assays. In addition, in some cases it was not possible to perform all possible analyses on every mouse run on a particular behavioral assay, e.g., because some analyses were performed at later times and the original data had not been recorded/stored in a manner that was suitable for a specific analysis. This explains why the *N*s sometimes differ for multiple analyses of data from the same assay. Importantly, no animals were excluded from specific analyses post-hoc.

## Elevated plus maze

Mice were exposed to the EPM for a single 15 min session. All mice were placed in the center of the maze facing an open arm. Time spent in zones, distance traveled, and number of entries were scored with ANY-maze; head dips were manually scored by a blinded observer.

## Social/novel assay

Mice were exposed to a conspecific juvenile followed by a novel object in their home cage for 10 min each. Active interaction time was scored by a blinded observer.

## Marble burying

Marble burying was performed as previously described (*Angoa-Pérez et al., 2013*). Mice were placed in a larger housing cage for 20 min with 20 marbles arranged in a 4 × 5 grid. After 20 min, the number of fully buried marbles was counted.

## Cognitive tasks

Mice were singly housed and placed on a reverse light-dark cycle for the duration of testing. Mice received 3 days of restricted food intake to reach a goal weight of ~80% free-feeding weight in order to sufficiently motivate them. In each task, this period was used to habituate mice to testing apparatus and basic task mechanics (location of food reward, trial structure, etc.). Water was freely available during the entire period. All testing was done under red light.

## Rule shifting

An odor/texture rule shifting task was performed as previously described (*Cho et al., 2015*; *Ellwood et al., 2017*). Briefly, mice were presented with two bowls containing either sand (Mosser Lee White Sand) or bicarbonate-free cat x (1% by volume) with either ground coriander (McCormick) or garlic powder (McCormick), as well as finely chopped peanut butter chips to mask scent of food reward. Each trial contained one of two possible combinations of media: sand and garlic paired with litter and coriander, respectively, or sand and coriander paired with litter and garlic, respectively. In the initial association phase of the task, mice had to learn that a single cue (e.g. sand) signaled the location of a reward. Once mice learned this rule (8 out of 10 previous trials correct), there was an un-cued extradimensional rule shift such that a different type of cue (e.g. garlic) now signaled the reward.

## Delayed match to sample task

A delayed match to sample T-maze task was performed as previously described (*Spellman et al., 2015*; *Tamura et al., 2017*). Briefly, mice were placed at the base of a T-shaped maze at the start of each trial. During the sample phase, one of the two choice arms of the T was blocked off such that mice were forced to one arm. After reaching the end of the arm, mice then had to return to the start point, where a sliding door held them for a variable delay phase (all data presented here from a 4 s delay). Following the delay was a choice phase – the door was removed, allowing the mice to run down the arms and choose which to enter. Mice had to learn to go to the opposite arm from the sample phase (e.g. if they entered the right arm during the sample phase, a food reward would be present in the left arm).

## Local field potential recordings

All surgeries were done under isofluorane anesthesia in a stereotaxic frame (Kopf). Standard-tip 0.5 MΩ-impedance stainless steel electrodes (Microprobes, SS30030.5A10) were inserted into the vHPC and mPFC. The coordinates for vHPC and mPFC were as follows: vHPC, −3.25 (AP), 3.1(ML), −4.1 (DV); mPFC, 1.7 (AP), 0.3 (ML), −2.75 (DV). A common reference screw was implanted into the cerebellum (0.5 mm posterior to lambda) and a silver ground wire was placed underneath the left lateral scalp. After affixing the electrodes in place using Metabond, connections were made to the headstage of a multi-channel recording system (Pinnacle). All channels shared a common reference (cerebellum). Data was collected at 2000 Hz and band-pass filtered 1–200 Hz at the pre-amp. Electrode placement was verified histologically. We also examined the power spectra from all electrodes; only animals with vHPC power spectra that exhibited a visible peak in the theta- frequency range as judged by a blinded observer were used for further analysis. 4 mice of each genotype were excluded due to lack of a visible theta peak (these mice were excluded from further workflow including histology).

Analysis of LFP data was facilitated using custom MATLAB code. The LFP signals were FIR-filtered (filter length 3x period corresponding to minimum frequency of frequency band) and Hilbert transformed to yield the instantaneous amplitudes (magnitude) and phases (angle). Bulk measures were calculated using data from the entire recording period; dynamic measures were calculated using a 2.5 s window, at 1.5 s intervals from 7.5 s before to 7.5 s after the animal entered the center of the EPM. Dynamic measurements were quantified as z-scores calculated relative to the rest of the run (7.5 s before to 7.5 s after the animal entered the center).

Power was quantified using Welch's power spectral density estimate with nonoverlapping segments. Synchrony between vHPC and mPFC was measured by taking the Hilbert transform of band-passed data and either comparing the instantaneous phase using the weighted-phase locking index (*Vinck et al., 2011*) or instantaneous amplitude using amplitude covariation. These measures were

computed across four frequency bands: theta (4–12 Hz), beta (13–30 Hz), low gamma (30–55 Hz), and high gamma (65–100 Hz).

Cross-frequency coupling was calculated by comparing the instantaneous phase in a low frequency band with the instantaneous amplitude in a high frequency band. Specifically, instantaneous phase and amplitude were obtained using the Hilbert transform (using the Matlab function *hilbert*). At each point in time, this phase and amplitude were combined to yield a vector in the complex plane. We combined vectors from successive timepoints, and the amplitude of the vector sum was normalized to the sum of all the amplitudes to quantify the strength of cross-frequency coupling. Low frequency bands were theta (2–6 Hz) and alpha (6–10 Hz). High frequency bands were beta (13–30 Hz), low gamma (30–55 Hz), and high gamma (65–100 Hz). Cross- frequency coupling was calculated for all possible combinations of a single low and single high frequency band in all combinations of brain regions (PFC low/HPC high, HPC high/PFC low, PFC low/PFC high, HPC low/HPC high).

These features (*Table 1*) were all used as input for the ICA based on methods outlined in previous work (*Kirkby et al., 2018*). First, all features were calculated for each subject and PCA was performed for dimensionality reduction and orthogonalization and the number of significant components was calculated using the threshold set by the Marchenko-Pastur Law (*Lopes-dos-Santos et al., 2013*). ICA was used on the significant PCs to separate the signal mixtures into independent sources using the *fastICA* algorithm (*Hyvärinen and Oja, 2000*). Similarity of ICs across mice was calculated using the Pearson correlation coefficient. Significant clusters were isolated by selecting for ICs that had a correlation coefficient of >0.7 with at least one other IC and using MATLAB's *graph* function to identify groups of highly similar ICs. Characteristic ICs were found by averaging groups of ICs with members from at least three different animals. The projection of these characteristic ICs onto behavior was found by multiplying the vector of Z-scored features in each point in time by the weight in the characteristic IC and summing all values.

## Whole cell patch clamp recordings

Mice were injected with 750 nL of AAV5-CaMKIIa-hChR2(H134R)-EYFP (UNC Vector Core) into the vHPC (DV: −4, AP: −3.3, ML: −3.2) to label excitatory projections from the vHPC to the mPFC. A subset of mice were also injected with 500 nL AAV-DlxI12b-mCherry in the mPFC (DV: −2.75, AP: 1.7, ML: 0.3) to label MGE-derived interneurons (*Potter et al., 2009*). We waited ~8 weeks from virus injection to slice experiments. Whole cell patch recordings were obtained from 250 µm coronal slices. Cells were identified using differential contrast video microscopy on an upright microscope (BX51W1, Olympus) and recordings were made using a Multiclamp 700A (Molecular Devices). Data was collected using pClamp (Molecular Devices) software and analyzed using custom MATLAB code. Patch electrodes were filled with the following (in mM): 130 K-gluconate, 10 KCl, 10 HEPES, 10 EGTA, 2 MgCl, 2 MgATP, and 0.3 NaGTP (pH adjusted to 7.3 with KOH). All recordings were at 32.0 ± 1˚C. Series resistance was usually 10–20 MΩ, and experiments were discontinued above 25 MΩ. For voltage clamp recordings, cells were held at −70 mV and +10 mV to isolate EPSCs and IPSCs, respectively. An LED engine (Lumencor) was used for optogenetic stimulation of terminals from vHPC projections. We used ~1–3 mW of 470 nm light in 5 ms pulses to stimulate ChR2-infected fibers. The light was delivered to the slice via a 40x objective (Olympus) which illuminated the full field.

## Computational model of the role of feedforward inhibition

The effects of changing the strength of excitatory drive onto interneurons was modeled using two integrate-and-fire neurons – an output cell, representing a pyramidal cell, and an interneuron that targeted the output cell, representing a FSIN. Each cell received noise input and theta-patterned 'hippocampal' input. Initial values were selected such that the inhibitory neuron would spike at ~20 Hz and the output neuron would spike at ~25 Hz and ~50 Hz in the presence and absence of inhibition. All values were held constant except for the strength of excitatory input onto the output-targeting interneuron, adjusting either just the hippocampal strength or adjusting the hippocampal and noise strength in parallel. Input spikes were modeled as a Poisson process. Correlation between the input sources was calculated by comparing binned spike times for input spikes (from the Poisson train) and output spikes (when the output cell's membrane potential cleared a threshold). The relative contributions of the two input sources was calculated by comparing the ratio of the correlation

**Table 3.** Details of all statistical tests N indicates biological replicates for example individual cells or behavior trials.

| Figure | Data | Test | P val | WT Animals | Het Animals | WT n | Het n |
|--------|------|------|-------|------------|-------------|------|-------|
| *Figure 1C* | Zone occupancy | Wilcoxon rank sum | 0.003 | 18 | 27 | | |
| *Figure 1D* | EPM Distance | Wilcoxon rank sum | 0.35 | 16 | 23 | | |
| *Figure 1E* | Open time | Wilcoxon rank sum | 0.001 | 18 | 27 | | |
| *Figure 1E* | Center time | Wilcoxon rank sum | 0.02 | 18 | 27 | | |
| *Figure 1F* | Head dips | Wilcoxon rank sum | 0.03 | 14 | 14 | | |
| *Figure 1G* | Open entries | Wilcoxon rank sum | 0.32 | 16 | 23 | | |
| *Figure 1H* | Open visit | Wilcoxon rank sum | 0.047 | 16 | 23 | | |
| *Figure 2B* | WPLI, t = 0 | Wilcoxon rank sum | 0.0007 | 6 | 7 | 274 | 316 |
| *Figure 2B* | WPLI, t = 1.5 | Wilcoxon rank sum | 0.043 | 6 | 7 | 274 | 316 |
| *Figure 2B* | WPLI, t = −3,–1.5, 0, +1.5 during closed-center runs | Linear mixed effects model timepoint mouse genotype timept X genotype | 0.0026 0.47 0.059 0.0004 | 6 | 7 | 274 | 316 |
| *Figure 2C* | Avg zone WPLI, genotype | Two-way ANOVA | 0.03 | 6 | 7 | | |
| *Figure 2C* | Avg zone WPLI, zone | Two-way ANOVA | 0.063 | 6 | 7 | | |
| *Figure 2C* | Avg zone WPLI, interaction | Two-way ANOVA | 0.98 | 6 | 7 | | |
| *Figure 2D* | Theta WPLI | Wilcoxon rank sum | 0.031 | 6 | 7 | | |
| *Figure 3E* | IC zone projection, t = 0 | Wilcoxon rank sum | 0.007 | 6 | 7 | 39 | 37 |
| *Figure 3E* | ICA zone projection t = 0 vs. baseline (average of first and last timepoints) during closed-center-open runs | Linear mixed effects model timepoint mouse genotype timept X genotype | 0.085 0.16 0.0044 0.010 | 6 | 7 | 39 | 37 |
| *Figure 4E* | FSIN charge | Wilcoxon rank sum | 0.006 | 6 | 3 | 11 | 7 |
| *Figure 4F* | FSIN PPR | Wilcoxon rank sum | 0.03 | 6 | 3 | 11 | 7 |
| *Figure 4G* | FSIN latency | Wilcoxon rank sum | 0.013 | 6 | 3 | 11 | 7 |
| *Figure 4H* | FSIN # spikes | Wilcoxon rank sum | 0.08 | 6 | 3 | 11 | 7 |
| *Figure 5E* | Pyr charge | Wilcoxon rank sum | 0.28 | 13 | 8 | 17 | 11 |
| *Figure 5F* | Pyr PPR | Wilcoxon rank sum | 0.15 | 13 | 8 | 17 | 11 |
| *Figure 5G* | Pyr latency | Wilcoxon rank sum | 0.76 | 13 | 8 | 17 | 11 |
| *Figure 5H* | Pyr # spikes | Wilcoxon rank sum | 0.78 | 13 | 8 | 17 | 11 |
| *Figure 1—figure supplement 1* | Sex-corrected zone occupancy | Wilcoxon rank sum | 0.013 | 18 | 27 | | |
| *Figure 1—figure supplement 1A* | Zone occupancy for Het M vs. F | Wilcoxon rank sum | 0.60 | | M: 10, F: 17 | | |
| *Figure 1—figure supplement 1B* | Sex-corrected EPM distance | Wilcoxon rank sum | 0.79 | 16 | 23 | | |
| *Figure 1—figure supplement 1C* | Head dips: genotype | 2-way ANOVA | 0.02 | M: 10, F: 4 | M: 8, F: 6 | | |
| *Figure 1—figure supplement 1C* | Head dips: sex | 2-way ANOVA | 0.81 | M: 10, F: 4 | M: 8, F: 6 | | |
| *Figure 1—figure supplement 1C* | Head dips: genotype X sex | 2-way ANOVA | 0.36 | M: 10, F: 4 | M: 8, F: 6 | | |
| *Figure 1—figure supplement 1C* | Head dips for Het M vs. F | Wilcoxon rank sum | 0.54 | | M: 8, F: 6 | | |
| *Figure 1—figure supplement 1D* | Open arm entries: genotype | 2-way ANOVA | 0.22 | M: 12, F: 4 | M: 9, F: 14 | | |
| *Figure 1—figure supplement 1D* | Open arm entries: sex | 2-way ANOVA | 0.61 | M: 12, F: 4 | M: 9, F: 14 | | |
| *Figure 1—figure supplement 1D* | Open entries: genotype X sex | 2-way ANOVA | 0.32 | M: 12, F: 4 | M: 9, F: 14 | | |

*Table 3 continued on next page*

*Table 3 continued*

| Figure | Data | Test | P val | WT Animals | Het Animals | WT n | Het n |
|--------|------|------|-------|------------|-------------|------|-------|
| *Figure 1—figure supplement 1E* | Open visit length: genotype | 2-way ANOVA | 0.22 | M: 12, F: 4 | M: 9, F: 14 | | |
| *Figure 1—figure supplement 1E* | Open visit length: sex | 2-way ANOVA | 0.49 | M: 12, F: 4 | M: 9, F: 14 | | |
| *Figure 1—figure supplement 1E* | Open visit length: genotype X sex | 2-way ANOVA | 0.76 | M: 12, F: 4 | M: 9, F: 14 | | |
| *Figure 1—figure supplement 1F* | Sex-corrected open arm time | Wilcoxon rank sum | 0.013 | 16 | 23 | | |
| *Figure 1—figure supplement 1F* | Open arm time: Het M vs. F | Wilcoxon rank sum | 0.61 | | M: 10, F: 17 | | |
| *Figure 1—figure supplement 1G* | Center time: genotype | 2-way ANOVA | 0.087 | M: 12, F: 6 | M: 10, F: 17 | | |
| *Figure 1—figure supplement 1G* | Center time: sex | 2-way ANOVA | 0.29 | M: 12, F: 6 | M: 10, F: 17 | | |
| *Figure 1—figure supplement 1G* | Center time: genotype X sex | 2-way ANOVA | 0.48 | M: 12, F: 6 | M: 10, F: 17 | | |
| *Figure 1—figure supplement 2A* | Social interaction | Wilcoxon rank sum | 0.34 | 7 | 7 | | |
| *Figure 1—figure supplement 2B* | Novel objection | Wilcoxon rank sum | 0.95 | 7 | 7 | | |
| *Figure 1—figure supplement 2C* | Marble burying | Wilcoxon rank sum | 0.45 | 8 | 7 | | |
| *Figure 1—figure supplement 2D* | OF distance | Wilcoxon rank sum | 0.15 | 14 | 17 | | |
| *Figure 1—figure supplement 2F* | T-maze trials | Wilcoxon rank sum | 0.6 | 5 | 5 | | |
| *Figure 1—figure supplement 2H* | Rule shift IA | Wilcoxon rank sum | 0.89 | 4 | 4 | | |
| *Figure 1—figure supplement 2H* | Rule shift RS | Wilcoxon rank sum | 0.89 | 4 | 4 | | |
| *Figure 2—figure supplement 1A* | PFC theta power | Wilcoxon rank sum | 0.91 | 6 | 7 | | |
| *Figure 2—figure supplement 1A* | PFC beta power | Wilcoxon rank sum | 0.94 | 6 | 7 | | |
| *Figure 2—figure supplement 1A* | PFC low gamma power | Wilcoxon rank sum | 0.47 | 6 | 7 | | |
| *Figure 2—figure supplement 1A* | PFC high gamma power | Wilcoxon rank sum | 0.8 | 6 | 7 | | |
| *Figure 2—figure supplement 1B* | HPC Theta power | Wilcoxon rank sum | 0.23 | 6 | 7 | | |
| *Figure 2—figure supplement 1B* | HPC Beta power | Wilcoxon rank sum | 0.093 | 6 | 7 | | |
| *Figure 2—figure supplement 1B* | HPC low gamma power | Wilcoxon rank sum | 0.17 | 6 | 7 | | |
| *Figure 2—figure supplement 1B* | HPC high gamma power | Wilcoxon rank sum | 0.94 | 6 | 7 | | |
| *Figure 2—figure supplement 1C* | PFC closed theta power | Wilcoxon rank sum | 0.88 | 6 | 7 | | |
| *Figure 2—figure supplement 1C* | PFC closed beta power | Wilcoxon rank sum | 1 | 6 | 7 | | |
| *Figure 2—figure supplement 1C* | PFC closed LG power | Wilcoxon rank sum | 0.29 | 6 | 7 | | |

*Table 3 continued*

| Figure | Data | Test | P val | WT Animals | Het Animals | WT n | Het n |
|--------|------|------|-------|------------|-------------|------|-------|
| *Figure 2—figure supplement 1C* | PFC closed HG power | Wilcoxon rank sum | 0.1 | 6 | 7 | | |
| *Figure 2—figure supplement 1D* | HPC closed theta power | Wilcoxon rank sum | 0.25 | 6 | 7 | | |
| *Figure 2—figure supplement 1D* | HPC closed beta power | Wilcoxon rank sum | 0.15 | 6 | 7 | | |
| *Figure 2—figure supplement 1D* | HPC closed LG power | Wilcoxon rank sum | 0.48 | 6 | 7 | | |
| *Figure 2—figure supplement 1D* | HPC closed HG power | Wilcoxon rank sum | 0.89 | 6 | 7 | | |
| *Figure 2—figure supplement 1E* | PFC open theta power | Wilcoxon rank sum | 0.89 | 6 | 7 | | |
| *Figure 2—figure supplement 1E* | PFC open beta power | Wilcoxon rank sum | 0.89 | 6 | 7 | | |
| *Figure 2—figure supplement 1E* | PFC open LG power | Wilcoxon rank sum | 1 | 6 | 7 | | |
| *Figure 2—figure supplement 1E* | PFC open HG power | Wilcoxon rank sum | 0.15 | 6 | 7 | | |
| *Figure 2—figure supplement 1F* | HPC open theta power | Wilcoxon rank sum | 0.32 | 6 | 7 | | |
| *Figure 2—figure supplement 1F* | HPC open beta power | Wilcoxon rank sum | 0.2 | 6 | 7 | | |
| *Figure 2—figure supplement 1F* | HPC open LG power | Wilcoxon rank sum | 0.25 | 6 | 7 | | |
| *Figure 2—figure supplement 1F* | HPC open HG power | Wilcoxon rank sum | 1 | 6 | 7 | | |
| *Figure 3—figure supplement 1D* | IC #1 zone projection | Wilcoxon rank sum | 0.007 | 6 | 7 | 39 | 37 |
| *Figure 3—figure supplement 1D* | IC #1 zone projection t = 0 vs. baseline (average of first and last timepoints) during closed-center-open runs | Linear mixed effects model timepoint mouse genotype timept X genotype | 0.085 0.16 0.0044 0.010 | 6 | 7 | 39 | 37 |
| *Figure 3—figure supplement 1F* | IC #3 zone projection | Wilcoxon rank sum | 0.015 | 6 | 7 | 39 | 37 |
| *Figure 3—figure supplement 1F* | IC #3 zone projection t = 0 vs. baseline (average of first and last timepoints) during closed-center-open runs | Linear mixed effects model timepoint mouse genotype timept X genotype | 0.0094 0.50 0.026 0.052 | 6 | 7 | 39 | 37 |
| *Figure 4—figure supplement 1C* | FSIN resting potential | Wilcoxon rank sum | 0.50 | 6 | 3 | 11 | 7 |
| *Figure 4—figure supplement 1D* | FSIN input resistance | Wilcoxon rank sum | 0.44 | 6 | 3 | 11 | 7 |
| *Figure 4—figure supplement 1E* | FSIN halfwidth | Wilcoxon rank sum | 0.47 | 6 | 3 | 11 | 7 |
| *Figure 4—figure supplement 1F* | FSIN max firing rate | Wilcoxon rank sum | 0.50 | 6 | 3 | 11 | 7 |
| *Figure 5—figure supplement 1C* | Pyr resting potential | Wilcoxon rank sum | 0.94 | 13 | 8 | 17 | 11 |
| *Figure 5—figure supplement 1D* | Pyr input resistance | Wilcoxon rank sum | 0.80 | 13 | 8 | 17 | 11 |
| *Figure 5—figure supplement 1E* | Pyr halfwidth | Wilcoxon rank sum | 0.46 | 13 | 8 | 17 | 11 |
| *Figure 5—figure supplement 1F* | Pyr max firing rate | Wilcoxon rank sum | 0.93 | 13 | 8 | 17 | 11 |

between the output spikes and the noise input or hippocampal input. Correlation values were based on 1000 iterations of a 1 s spike train.

## Statistics, data analysis, and data and code availability

Unless otherwise specified, non-parametric tests were used for all statistical comparisons and all tests are two-sided. Statistics were calculated using MATLAB or Python's SciPy package. Linear mixed models were evaluated using the 'fitlme' function in Matlab. Sample sizes were based on prior studies. All Ns indicate biological replication, that is, data from different samples (different cells or different animals), rather than technical replication (multiple measurements of the same sample). Details of p-values, Ns and statistical tests for all comparisons performed in this study are given in *Table 3*. Raw data related to this study has been deposited in Dryad (doi:10.7272/Q6ZP44B9). All custom written analysis code is available on Github (*Cunniff, 2020*) (https://github.com/mcunniff/PogZ_paper); *Cunniff, 2020* copy archived at swh:1:rev:189f9c500bdeaddeb69d3eef8b604949c2936d19.

## Acknowledgements

We acknowledge funding from the Simons Foundation Autism Research Initiative (Grants # 399853 and 514438), NIMH (R56MH117961 and R01MH117961), and a Trailblazer Award from the Weill Institute for Neurosciences.

## Additional information

### Competing interests

John LR Rubenstein: JLRR is cofounder, stockholder, and currently on the scientific board of Neurona, a company studying the potential therapeutic use of interneuron transplantation. Vikaas Singh Sohal: VSS has received research funding from Neurona therapeutics and is on the scientific board of Empathic Therapeutics. The other authors declare that no competing interests exist.

### Funding

| Funder | Grant reference number | Author |
| --- | --- | --- |
| Simons Foundation | 399853 | Margaret M Cunniff<br>Vikaas Singh Sohal |
| National Institute of Mental Health | R56MH117961 | Margaret M Cunniff<br>Vikaas Singh Sohal |
| Weill Institute for Neurosciences | | Margaret M Cunniff<br>Vikaas Singh Sohal |
| Simons Foundation | 514438 | Eirene Markenscoff-Papadimitriou<br>John LR Rubenstein |
| National Institute of Mental Health | R01MH117961 | Margaret M Cunniff<br>Vikaas Singh Sohal |

The funders had no role in study design, data collection and interpretation, or the decision to submit the work for publication.

### Author contributions

Margaret M Cunniff, Conceptualization, Formal analysis, Investigation, Visualization, Writing - original draft; Eirene Markenscoff-Papadimitriou, Resources, Methodology; Julia Ostrowski, Software, Formal analysis, Visualization; John LR Rubenstein, Resources, Supervision, Funding acquisition, Project administration, Writing - review and editing; Vikaas Singh Sohal, Conceptualization, Software, Supervision, Funding acquisition, Methodology, Project administration, Writing - review and editing

### Author ORCIDs

John LR Rubenstein http://orcid.org/0000-0002-4414-7667
Vikaas Singh Sohal https://orcid.org/0000-0002-2238-4186

### Ethics

Animal experimentation: All experiments were performed in strict accordance with the recommendations in the Guide for the Care and Use of Laboratory Animals of the National Institutes of Health. All of the animals were handled according to approved institutional animal care and use committee (IACUC) protocols of the University of California, San Francisco (IACUC protocol #AN170116).

### Decision letter and Author response

Decision letter https://doi.org/10.7554/eLife.54835.sa1
Author response https://doi.org/10.7554/eLife.54835.sa2

## Additional files

### Supplementary files

• Transparent reporting form

### Data availability

All data has been deposited in Dryad, DOI: doi:10.7272/Q6ZP44B9 All code has been deposited in GitHub: https://github.com/mcunniff/PogZ_paper (copy archived at https://archive.softwareheritage.org/swh:1:rev:189f9c500bdeaddeb69d3eef8b604949c2936d19/).

The following dataset was generated:

| Author(s) | Year | Dataset title | Dataset URL | Database and Identifier |
|---|---|---|---|---|
| Sohal VS | 2020 | Altered hippocampal-prefrontal communication during anxiety-related avoidance in mice deficient for the autism-associated gene PogZ | http://dx.doi.org/10.5061/dryad.Q6ZP44B9 | Dryad Digital Repository, 10.5061/dryad.Q6ZP44B9 |

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
