## [Decision Letter]

**Acceptance summary:**

The authors find that mice deficient for the ASD-linked gene POGZ show decreased anxiety, impaired theta synchrony during anxiety assays, and decreased hippocampal input to mPFC cortical fast-spiking cells. These findings will be of interest to those working as autism, as POGZ is a high confidence autism gene. In combination with other similar studies, these results could provide a means for assessing whether potential new treatments can alleviate deficits in animal models.

**Decision letter after peer review:**

Thank you for submitting your article "Altered hippocampal-prefrontal communication and anxiety-related behavior in mice deficient for the ASD-linked gene POGZ" for consideration by *eLife*. Your article has been reviewed by three peer reviewers, and the evaluation has been overseen by Laura Colgin as the Reviewing and Senior Editor. The following individual involved in review of your submission has agreed to reveal their identity: Alex Harris (Reviewer #3).

The reviewers have discussed the reviews with one another and the Reviewing Editor has drafted this decision to help you prepare a revised submission.

In the latter case, our expectation is that the authors will eventually carry out the additional experiments and report on how they affect the relevant conclusions either in a preprint on bioRxiv or medRxiv, or if appropriate, as a Research Advance in *eLife*, either of which would be linked to the original paper. If the claims in question are instead removed, additional experiments would not be expected.

Summary:

Cunniff et al. provide a comprehensive account of vHPC/mPFC-dependent behavior and neurophysiological abnormalities in mice harboring heterozygous loss-of-function mutations in the ASD-associated *Pogz* chromatin remodelling gene (*Pogz^+/-^* mice). Their interpretation of results is supported using a computational model, predicting that reduced excitatory drive onto mPFC FSINs decreases the correlation between mPFC pyramidal cell output and hippocampal input. The impressive breadth of techniques constitutes a logical and complementary approach to delineating vHPC-mPFC abnormalities in *Pogz* mice.

However, reviewers agreed that some of the major claims made by the authors were not well supported by their data. Moreover, reviewers agreed that it is unclear whether the deficits these *Pogz* mice display are relevant to autism and felt that the translational claims made in the paper should be tempered. Additionally, there are some few presentation issues that need to be addressed.

Reviewers felt that the authors cannot claim any specificity of the decreased hippocampal input to mPFC cortical fast-spiking cells for the reported behavioral and theta synchrony findings, as one would intuit brain-wide changes as a result of disrupting a gene implicated in such universally important processes. Also, the translational claims and links to autism were viewed as overstated. There were other presentation and analysis issues that should also be addressed. See Major Comments below for details.

Revisions expected in follow-up work:

If authors choose to keep the overstated causal relationship between physiological and behavioral findings in the paper, and the unsupported specificity claims, then appropriate rescue experiments would be expected in the future.

Major Comments:

1) The authors find that *Pogz^+/-^* mice show: decreased anxiety, impaired theta synchrony during anxiety assays and decreased hippocampal input to mPFC cortical fast-spiking cells. Overall, these findings are interesting and relevant to autism, as *Pogz* is a high confidence autism gene. The three findings may interact with each other to produce the anxiety behavioral deficit *Pogz* mice show. That is the narrative the authors favor. However, it is important to consider that as the authors state, *Pogz* "is known to play a role in chromatin regulation, mitotic progression, and chromosome segregation ". These are ubiquitous processes that likely affect a myriad of cell types and countless circuits in the brain. Thus, from the data shown, it is not possible to support the conclusions the authors claim.

For example, the theta synchrony deficit may be related to altered hippocampal input to cortical interneurons as the authors state. But it is also entirely possible that the observed theta synchrony deficit is due to changes in some neuromodulatory input to cortex in *Pogz* mice, or due to some other change induced by *Pogz*. The point is that *Pogz* affects so many developmental processes that it is not possible to state with any reasonable confidence that input to fast spiking interneurons causes the observed that synchrony deficit. If *Pogz* only selectively affected the development of hippocampal input to cortical interneurons, then the authors would be justified in their claims. Similarly, the behavioral deficit may be related to alterations in theta synchrony, but it may also be due to some other completely unrelated function of *Pogz* during development.

To truly support the claims, the authors would have to show that somehow rescuing or normalizing hippocampal input to fast spiking interneurons would both normalize theta synchrony deficits and behavioral symptoms displayed by *Pogz* mice. As it is not possible to perform this experiment, the authors should explicitly and clearly write that their findings provide a plausible, possible explanation for the behavioral deficit. But these findings do not prove that hippocampal inputs to fast spiking interneurons in cortex cause lower anxiety in *Pogz* mice. Unsubstantiated claims such as "We found deficient theta-frequency synchronization between the vHPC and mPFC in vivo. Furthermore, this involves a specific deficit in excitatory input from vHPC onto prefrontal GABAergic interneurons" should be removed from the paper. The authors do not show that the theta synchronization deficit involves a deficit in vhpc input to interneurons. They only show these two deficits exist, not that one is involved in the occurrence of the other. Similarly, the claim "we are able to show that the theta coordination of vHPC-mPFC activity and open arm avoidance can be disrupted simply by suppressing vHPC input to interneurons" is unsupported. The authors did not show a mechanistic connection between their 3 main findings. They only show 3 differences between *Pogz* and WT mice. On a positive note, later the authors state "we found reduced excitatory drive from vHPC onto fast-spiking interneurons. This synaptic abnormality could plausibly contribute to the abnormalities we found in both avoidance behavior and LFP synchrony". This is the appropriate interpretation of the results, and this is the tone that should be used throughout the paper.

The alternative argument that *Pogz* functions as a tool to study selective disruption of excitatory input to PFC interneurons is intriguing but would change the focus of the paper to establishing that claim by providing a) the mechanism for such selectivity and b) demonstrating that it exists in vivo.

2) Of note, a very recent Matsumura et al., 2020, paper generates a different line of *Pogz^+/-^* mice, also reporting that these mice spend more time in the central zone of an open field, alongside cortical thinning and hyperactivation of anterior cingulate cortex during social behavior. Importantly, Matsumara et al. rescue abnormalities using the AMPA receptor negative allosteric modulator perampanel. It would be of great interest to see if this same approach could alleviate the deficits in Cunniff et al.'s model. Reference to the Matsumura et al., 2020, and relevant comparisons with the present work will have to be added in Introduction and Discussion.

3) The authors should delete or tone down the "Clinical and therapeutical implications" section. It is unfeasible at this point to begin to plan to implement a therapy for autism based on these findings. There are no known methods that can selectively control hippocampal inputs to fast spiking cortical neurons in human patients. The authors summarize this state of affairs as "it is not immediately obvious how one would translate our findings into new treatments", making this whole section highly speculative.

4) Related to the above points, face validity may be an imperfect metric of translational utility, but that doesn't mean that the disruption of any behavior involving the prefrontal cortex has relevance for autism/intellectual disability. An alternative explanation is that the gene functions differently in mice and people. This is particularly the case since tests for the expected deficits (social and intellectual functioning) appeared intact. However, it may be premature to make this claim: these latter tests are underpowered (1/3 to 1/2 of the N used for EPM), and a recent paper (Matsumura et al., 2020) found social and cognitive deficits in *Pogz* missense mutation mice. To be translationally relevant, the authors would need to show that the long-range synchrony deficits underlie social or cognitive behavioral deficits. The *Pogz* mice in this study show only a very selective behavioral deficit in lower anxiety, which is not a typical autism symptom. These data raise questions on the validity of using *Pogz* mice for modeling autism phenotypes. This issue must be discussed clearly in the Discussion.

5) Materials and methods state that LFP electrode placements were checked histologically. Were any mice excluded on these grounds? It becomes important to show these data in light of a related recent paper by Matsumura et al., 2020, showing smaller brains in a different line of *Pogz^+/-^* mice. It is therefore feasible that electrode placement was systematically different between WT and+/- mice, contributing to WPLI and other neurophysiological differences.

6) How is it possible to have hundreds of biological replicates of the weighted phase locking measure with only ~20 mice?

7) Given that WPLI was reduced in both homecage and in EPM, presumably it is also reduced on the T-maze, yet *Pogz^+/-^* delayed alternation was apparently normal. Why is this, in light of all the work from the Gordon lab? Why would the phenotype preferentially manifest on the EPM?

8) Figure 3: Why is this particular IC highlighted? (i) How is the appropriate threshold for significant ICs calculated? (ii) How do the significant ICs in 3C relate to the points in the clusters in 3D? (iii) What is this justification for focus on the cluster that happens to correspond to theta-related to cross-frequency (phase-amplitude) coupling? And (iv) how does this relate to theta WPLI?

9) The authors argue based on their in vitro data and modeling that decreased vHPC excitatory input to FSIN are responsible for the *Pogz* deficits in theta synchrony. However, this model would predict mPFC spike phase-locking with vHPC theta (which they don't show) not theta phase synchrony or cross-frequency coupling. They should either show decreased spike phase locking or a model that accounts for their synchrony observations. It would be powerful to use the computational model to explore different frequency ranges of HPC input: does the model predict specific effects on theta-frequency inputs?

10) Does homecage WPLI predict EPM behavior in individual animals?

11) This model is proposed as an alternative to the suggestion that vHPC provides direct excitatory input to pyramidal neurons, but it doesn't address Padilla-Coreano et al., 2016 findings that blocking excitatory input did not affect overall firing rates of either pyramidal or interneurons but decreased firing rates in the preferred arm. By their model, shouldn't firing rates increase in the non-preferred arm?

12) Figure 2: Was the directionality of the HPC-PFC WPLI measure consistent with an effect on HPC drive to PFC?

13) Figure 3 needs to be expanded and clarified:

a) How was cross-frequency coupling calculated? The Materials and methods state that "Cross-frequency coupling was calculated by comparing the instantaneous phase in a high frequency band with the instantaneous amplitude in a low frequency band". Usually the converse would be true, but still more details are required – was an established method like the Modulation Index used?

b) Although it is possible to understand the results in Figure 3, it is not very reader-friendly. The authors should add some panels explaining each step in this analysis, showing visually how they go from raw LFP recordings to the plots shown in Figure 3.

c) There is insufficient information given about the empirical salient feature analysis. Figures 3A-D are not labeled in a way that readers can decipher the identities of features. The authors should show how all the features fared throughout the pipeline. Moreover, why doesn't this data analysis strategy pull out the known finding (theta phase synchrony)?

[Editors' note: further revisions were suggested prior to acceptance, as described below.]

Thank you for submitting your article "Altered hippocampal-prefrontal communication and anxiety-related behavior in mice deficient for the ASD-linked gene POGZ" for consideration by *eLife*. Your revised article has been reviewed by three peer reviewers, and the evaluation has been overseen by Laura Colgin as the Senior Editor. The following individual involved in review of your submission has agreed to reveal their identity: Alex Harris (Reviewer #3).

The reviewers have discussed the reviews with one another and the Reviewing Editor has drafted this decision to help you prepare a revised submission.

As the editors have judged that your manuscript is of interest, but as described below that additional analyses are required before a final decision can be made.

We would like to draw your attention to changes in our revision policy that we have made in response to COVID-19 (https://elifesciences.org/articles/57162). First, because many researchers have temporarily lost access to the labs, we will give authors as much time as they need to submit revised manuscripts. We are also offering, if you choose, to post the manuscript to bioRxiv (if it is not already there) along with this decision letter and a formal designation that the manuscript is "in revision at *eLife*". Please let us know if you would like to pursue this option. (If your work is more suitable for medRxiv, you will need to post the preprint yourself, as the mechanisms for us to do so are still in development.)

Summary:

Many positive points were noted about the revised submission. This resubmission addressed key concerns by reducing claims of causality and translational relevance. The authors make a reasonable case justifying the relevance of their behavioral and circuit finding. The authors give a reasonable explanation for why delayed alternation might be spared in the T-maze. They have done analyses to address the impact of homecage WLPI and sex on their findings. They analyze the directionality of the synchrony effect. They have added new modeling showing frequency specific effects. They have rewritten the text to clarify several of the points. However, some points were incompletely addressed, and some new concerns were raised by the responses to the first set of critiques. Reviewers agreed that robust evidence for the paper's conclusions requires additional analyses, and thus the ultimate decision on the manuscript will depend on the outcome of these analyses.

Essential revisions:

1) Even though it occurred in equal numbers across genotype, a concern was raised that in a paper focused on theta, 38% (8/21) of the mice were excluded for lacking a clear theta peak (despite having good histology). Is there any explanation for why theta is aberrant in such a large fraction of the data? What would happen to the findings if these mice were included? The authors should include any mice that don't have a technical problem with the electrodes or location. An alternative approach to point 1 would be to analyze the 4+4 “non theta” mice separately in a supplement.

2) In general, it seems that people tend to treat units recorded from the same animal as independent but consider LFP data recorded with the same electrode as a single biological replicate (and trials as experimental replicates). The authors should use a repeated measures design, use the average LFP per mouse, or some other statistically valid approach for their analyses.

3) The explanation of Figure 3 is improved, but questions remain about the procedure. It seems that ICs generated from a combination of WT and POGZ mice (should the number of mice be 13?). How was consistency across mice measured? By pairwise correlations? How did this procedure measure consistency across all mice? Was there any consideration given to whether ICs were consistent within or across genotype?

4) What percent of high correlation ICs contained a measure of theta synchrony? Similarly, what percent of ICs were task-modulated? Without these measures, it is hard to interpret the significance of a given IC's task modulation and genotype disruption.

5) The explanation of why ICA did not yield theta phase synchrony was unclear. Isn't theta synchrony a stable “network” conserved across mice? Doesn't that serve as an important positive control of this empiric method?

[Editors' note: further revisions were suggested prior to acceptance, as described below.]

Thank you for resubmitting your article "Altered hippocampal-prefrontal communication and anxiety-related behavior in mice deficient for the ASD-linked gene POGZ" for consideration by *eLife*. Your revised article has been reviewed by two peer reviewers, and the evaluation has been overseen by Laura Colgin as the Senior Editor and Reviewing Editor. The following individual involved in review of your submission has agreed to reveal their identity: Alex Harris (Reviewer #3).

The reviewers have discussed the reviews with one another and the Reviewing Editor has drafted this decision to help you prepare a revised submission.

Summary:

The authors have adequately addressed the discussion points. However, regarding the results, reviewers remain concerned about the analysis and agree that analysis concerns that were raised previously were not adequately addressed in the revised manuscript. Revisions that are considered essential for acceptance are explained below.

Essential revisions:

1) Reviewers remain concerned about the authors' decision to ignore the recommendation to use a repeated measures analysis for some of the reported results, in which different trials from the same mouse were treated as independent. Based on the statistics currently used, it is not convincing to draw the conclusion that theta band WLPI is task-modulated in WT, but not *PogZ^+/-^* mice. If there happened to be a dependency of the trials due to the mouse, then z-scoring wouldn't remove it. In other words, there could be some unknown reason why impairments are strong in some animals and not others (e.g., some interaction between the genotype and individual experiences during development), and z-scoring would not take care of that. Reviewers agree that a repeated measures analysis should be used. Also, for the effects in some frequency bands and not others, again, a repeated measures analysis would be best, as power in different frequency bands within the same recordings is also not independent. It is important to remember that, at *eLife*, the decision letter containing the consensus review and recommendations is published along with the article. So, we think that readers would wonder why authors did not follow the suggestion to apply a repeated measures analysis, and this could lower confidence in the results and diminish the paper's impact.

2) The authors should also include the exact statistics and degrees of freedom, APA style, not just p-values when reporting effects. Also, reviewers think it would be best to request that all the figure legends list N, name of statistical test, and p value, so that it is easier for readers to understand how data were analyzed. Currently some figures (such as Figure 1) list only p values.

3) The final N for Figure 3E and Figure 3—figure supplement 1D-F remains unclear. Are these N also based on z-scored trial data? If so, they are subject to the same statistical concern mentioned above.

---

## [Author Response]

Major Comments:1) The authors find that Pogz^+/-^ mice show: decreased anxiety, impaired theta synchrony during anxiety assays and decreased hippocampal input to mPFC cortical fast-spiking cells. Overall, these findings are interesting and relevant to autism, as Pogz is a high confidence autism gene. The three findings may interact with each other to produce the anxiety behavioral deficit Pogz mice show. That is the narrative the authors favor. However, it is important to consider that as the authors state, Pogz "is known to play a role in chromatin regulation, mitotic progression, and chromosome segregation ". These are ubiquitous processes that likely affect a myriad of cell types and countless circuits in the brain. Thus, from the data shown, it is not possible to support the conclusions the authors claim.For example, the theta synchrony deficit may be related to altered hippocampal input to cortical interneurons as the authors state. But it is also entirely possible that the observed theta synchrony deficit is due to changes in some neuromodulatory input to cortex in Pogz mice, or due to some other change induced by Pogz. The point is that Pogz affects so many developmental processes that it is not possible to state with any reasonable confidence that input to fast spiking interneurons causes the observed that synchrony deficit. If Pogz only selectively affected the development of hippocampal input to cortical interneurons, then the authors would be justified in their claims. Similarly, the behavioral deficit may be related to alterations in theta synchrony, but it may also be due to some other completely unrelated function of Pogz during development.To truly support the claims, the authors would have to show that somehow rescuing or normalizing hippocampal input to fast spiking interneurons would both normalize theta synchrony deficits and behavioral symptoms displayed by Pogz mice. As it is not possible to perform this experiment, the authors should explicitly and clearly write that their findings provide a plausible, possible explanation for the behavioral deficit. But these findings do not prove that hippocampal inputs to fast spiking interneurons in cortex cause lower anxiety in Pogz mice. Unsubstantiated claims such as "We found deficient theta-frequency synchronization between the vHPC and mPFC in vivo. Furthermore, this involves a specific deficit in excitatory input from vHPC onto prefrontal GABAergic interneurons" should be removed from the paper. The authors do not show that the theta synchronization deficit involves a deficit in vhpc input to interneurons. They only show these two deficits exist, not that one is involved in the occurrence of the other. Similarly, the claim "we are able to show that the theta coordination of vHPC-mPFC activity and open arm avoidance can be disrupted simply by suppressing vHPC input to interneurons" is unsupported. The authors did not show a mechanistic connection between their 3 main findings. They only show 3 differences between Pogz and WT mice. On a positive note, later the authors state "we found reduced excitatory drive from vHPC onto fast-spiking interneurons. This synaptic abnormality could plausibly contribute to the abnormalities we found in both avoidance behavior and LFP synchrony". This is the appropriate interpretation of the results, and this is the tone that should be used throughout the paper.

We agree with the point made by the Reviewers: we did not directly test causal relationships between observed physiological and behavioral abnormalities. Rather, the physiological abnormalities occur in a pathway that is known to influence the behaviors we measured, and thus constitute a plausible mechanism underlying behavioral abnormalities. However, Pogz influences many developmental processes, so we cannot rule out that Pogz may elicit behavioral abnormalities through alterations in other processes which we have not examined. We had tried to be circumspect about our claims in the original manuscript, but the reviewers’ feedback makes clear that in many cases the exact meaning of our text was ambiguous. Therefore, we have followed the reviewers’ recommendation to revise our text to make the nature of our claims more clear.

Specifically, we revised the last three sentences of the Abstract as follows:

“We found deficient theta-frequency synchronization between the vHPC and mPFC in vivo. When we examined vHPC-mPFC communication at higher resolution, vHPC input onto prefrontal GABAergic interneurons was specifically disrupted, whereas input onto pyramidal neurons remained intact. These findings illustrate how the loss of a high confidence autism gene can impair long-range communication by causing inhibitory circuit dysfunction within pathways important for specific behaviors.”

The alternative argument that Pogz functions as a tool to study selective disruption of excitatory input to PFC interneurons is intriguing but would change the focus of the paper to establishing that claim by providing a) the mechanism for such selectivity and b) demonstrating that it exists in vivo.2) Of note, a very recent Matsumura et al., 2020, paper generates a different line of Pogz^+/-^ mice, also reporting that these mice spend more time in the central zone of an open field, alongside cortical thinning and hyperactivation of anterior cingulate cortex during social behavior. Importantly, Matsumara et al. rescue abnormalities using the AMPA receptor negative allosteric modulator perampanel. It would be of great interest to see if this same approach could alleviate the deficits in Cunniff et al.'s model. Reference to the Matsumura et al,. 2020, and relevant comparisons with the present work will have to be added in Introduction and Discussion.

We have added a sentence in the Introduction describing key similarities between that study and ours, and a paragraph to the Discussion more thoroughly describing the Matsumara study.

3) The authors should delete or tone down the "Clinical and therapeutical implications" section. It is unfeasible at this point to begin to plan to implement a therapy for autism based on these findings. There are no known methods that can selectively control hippocampal inputs to fast spiking cortical neurons in human patients. The authors summarize this state of affairs as "it is not immediately obvious how one would translate our findings into new treatments", making this whole section highly speculative.

We have removed the “Clinical and therapeutic implications” section from the Discussion.

4) Related to the above points, face validity may be an imperfect metric of translational utility, but that doesn't mean that the disruption of any behavior involving the prefrontal cortex has relevance for autism/intellectual disability. An alternative explanation is that the gene functions differently in mice and people. This is particularly the case since tests for the expected deficits (social and intellectual functioning) appeared intact. However, it may be premature to make this claim: these latter tests are underpowered (1/3 to 1/2 of the N used for EPM), and a recent paper (Matsumura et al., 2020) found social and cognitive deficits in Pogz missense mutation mice. To be translationally relevant, the authors would need to show that the long-range synchrony deficits underlie social or cognitive behavioral deficits. The Pogz mice in this study show only a very selective behavioral deficit in lower anxiety, which is not a typical autism symptom. These data raise questions on the validity of using Pogz mice for modeling autism phenotypes. This issue must be discussed clearly in the Discussion.

We agree with the basic premise raised by the reviewers: simply finding a behavior that is abnormal when an autism gene is disrupted in mice does not mean that the underlying mechanisms will be relevant to autism. We also agree that face validity is not completely useless. However, there are two issues with face validity. First, social deficits in autism are complex and heterogenous, whereas mouse assays measure only the most rudimentary aspects of social behavior – social preference and/or preference for social novelty. In many individuals with autism, social preference and preference for social novelty are intact, but social functioning is disrupted in other ways. In particular, the largest study of individuals with disruptions in POGZ (25 individuals) found “in many cases, a seemingly contrary overly social and overly friendly demeanor” (Stessman et al., 2016). Furthermore, disruptions in genes of large effect size (like POGZ) elicit heterogeneous behavioral phenotypes. Thus, while lower anxiety may not be a general feature of autism, it may be relevant to individuals with disruptions in POGZ, because they have unusual phenotypes, including aggression, self-injury and irritability (White et al., 2016).

For the reasons outlined above, existing assays for mouse social behavior do not seem to possess even face validity for the unusual phenotypes found in patients with disruptions in POGZ. In this context, how should we study mechanisms that are likely to be relevant to the loss of POGZ? A logical approach is focusing on brain regions and networks that are broadly implicated in autism. While specific behaviors, e.g., social approach, may not be well conserved across species, there is a widespread belief that understanding general principles underlying the function of specific limbic circuits, e.g., interactions between the hippocampus and prefrontal cortex, will yield insights that translate across species. We are not asserting that what we found is definitively relevant to clinical autism. Rather we are arguing that studies of autism pathophysiology using mouse models should be driven by circuits, not behaviors. In this context, it would be premature to assert that social deficits are more likely to be relevant to autism than the phenotype we found – an inability of the PFC to use input to appropriately guide approach/avoidance decisions. Indeed, one hypothesis is that deficits in the ability of the PFC to appropriate guide approach/avoidance decisions is at the core of autism (Pfaff and Barbas, 2019). We have added a discussion of these issues to the Discussion under “Possible relevance of Pogz behavioral phenotypes to autism” (this replaces the “Clinical and therapeutic implications” section which has been removed).

Re: the comment that our studies of social behavior may have been underpowered. The design for our behavioral studies was to perform an initial screen using a large number of assays. This initial screen showed altered behavior in the EPM, but not in other behavioral assays, e.g., for social interaction. Then we validated the EPM deficit using additional mice. This is why the N is larger for the EPM than for other social and cognitive assays. We have added a note about this design to the Materials and methods to make this clear.

5) Materials and methods state that LFP electrode placements were checked histologically. Were any mice excluded on these grounds? It becomes important to show these data in light of a related recent paper by Matsumura et al., 2020, showing smaller brains in a different line of Pogz^+/-^ mice. It is therefore feasible that electrode placement was systematically different between WT and+/- mice, contributing to WPLI and other neurophysiological differences.

We verified electrode placement both histologically (by visually examining the anatomical location of the electrode track) and electrophysiologically (by confirming the presence of a prominent theta frequency peak in the LFP power spectrum). Importantly, as noted before, the fraction of experiments excluded due to the absence of a clear theta-frequency peak, was not different between WT and mutant mice (5 mice of each genotype were excluded) suggesting there was not systemic mistargeting in Pogz mutant mice as a result of anatomical differences. We have added text to the Discussion to acknowledge this possibility and discuss the reasons why we do not believe it was the case.

6) How is it possible to have hundreds of biological replicates of the weighted phase locking measure with only ~20 mice?

Table 3 specifies the Ns for this measurement – there were 274 closed arm-center runs from 6 WT mice, and 316 closed arm-center runs from 7 mutant mice. Each datapoint was converted to a z-score using the other points from the same run. This means the z-scored value on each run was statistically independent of z-scored values on other runs from the same mouse. We apologize that this was not clear in the original manuscript and have added text to the Materials and methods to make this clear.

7) Given that WPLI was reduced in both homecage and in EPM, presumably it is also reduced on the T-maze, yet Pogz^+/-^ delayed alternation was apparently normal. Why is this, in light of all the work from the Gordon lab? Why would the phenotype preferentially manifest on the EPM?

We agree that the physiological deficits we found are likely to impact other behaviors that also depend on hippocampal-prefrontal communication. So why didn’t we observe deficits in delayed alternation? There are three possible reasons. First, delayed alternation is much simpler than true T-maze based working memory tasks because the delay is much shorter (only 4 sec). Previous studies have shown that performance at such short delays does not require the same circuitry as longer delays (Bolkan et al., 2017). Second, the Gordon lab has previously shown how compensation can enable mutant mice to perform normally during spatially working memory tasks (Tamura et al., 2017). Specifically, other forms of synchronization may enable *Pogz* mutants to compensate for reduced theta-frequency phase synchronization. Third, we found that feedforward inhibition from the vHPC to mPFC is deficient. However, feedforward inhibition is mediated by multiple prefrontal cell-types including somatostatin (SST) and parvalbumin (PV) interneurons. The Gordon lab has recently shown that in mPFC, SST but not PV interneurons are required for spatial working memory (Abbas et al., 2018). Thus, it is possible that the deficits in feedforward inhibition we found spare some aspects of inhibition, and thus do not deleteriously impact spatial working memory. We have added a short paragraph discussing these possibilities in the Discussion.

8) Figure 3: Why is this particular IC highlighted? (i) How is the appropriate threshold for significant ICs calculated? (ii) How do the significant ICs in 3C relate to the points in the clusters in 3D? (iii) What is this justification for focus on the cluster that happens to correspond to theta-related to cross-frequency (phase-amplitude) coupling? And (iv) how does this relate to theta WPLI?

i) We compared the eigenvalues obtained using principal components analysis (PCA) to the Marchenko-Pastur distribution. The Marchenko-Pastur distribution yields the magnitude of eigenvalues expected by chance. Thus, the number of eigenvalues which exceed this threshold can be used to compute the number of significant dimensions. We then ran ICA to find this number of ICs.

ii) For the original figure all the significant ICs in panel 3C which are correlated with other ICs are shown in panel 3D (ICs which were not correlated with other ICs were omitted/not shown). Note: based on reviewer feedback, this figure has now been significantly altered (see comment 13).

iii-iv) We focused on this particular IC for two reasons. First, it measures the synchronization of theta-frequency hippocampal activity with activity in the prefrontal cortex. Thus, it measures theta-frequency communication/interaction between these two structures. Second, this IC exhibits modulation as mice approach the decision points in the EPM (namely the center zone). Thus, this IC represents a data-driven metric (i.e., a quantity discovered by unsupervised methods) that shows how theta-frequency communication between the vHPC and mPFC (phase-amplitude coupling between mPFC gamma and vHPC theta) correlates with approach-avoidance decisions (timepoints when the mouse enters the center zone). This makes it an almost perfect analog of theta-frequency WPLI, however, it was discovered in a data-driven manner rather than chosen a priori. Finding that this metric, like theta-frequency WPLI, is altered in *Pogz*^+/-^ mice during closed arm-center zone transitions thus provides strong confirmation that theta-frequency hippocampal-prefrontal communication related to approach-avoidance decisions is disrupted in *Pogz*^+/-^ mice.

9) The authors argue based on their in vitro data and modeling that decreased vHPC excitatory input to FSIN are responsible for the Pogz deficits in theta synchrony. However, this model would predict mPFC spike phase-locking with vHPC theta (which they don't show) not theta phase synchrony or cross-frequency coupling. They should either show decreased spike phase locking or a model that accounts for their synchrony observations. It would be powerful to use the computational model to explore different frequency ranges of HPC input: does the model predict specific effects on theta-frequency inputs?

First, as requested, we have simulated the responses of the model for different frequencies of HPC input. As now shown in Figure 6—figure supplement 2, our finding that feedforward inhibition enhances the signal-to-noise ration is frequency-specific. In particular, SNR is enhanced for delta and theta frequency inputs, but not for higher or lower frequency inputs.

We agree that decreased vHPC excitatory input to FSINs should decrease spike phase-locking with vHPC theta (at least for FSINs) as stated by the reviewers. However, we disagree with the next part of this statement. Specifically, LFP synchrony (either theta phase synchrony or cross frequency coupling) between the vHPC and mPFC reflects synchronization, not of spiking, but rather of local fields which are driven largely by synaptic currents (Buzsáki, Anastassiou and Koch, 2012; Haider et al., 2016). This is because synaptic currents are more extensive in both space (e.g., extending over larger membrane surface areas) and time compared to spikes. In the awake cortex, local field potentials seem to be particularly dominated by inhibitory activity (Teleńczuk et al., 2017). Reduced feedforward drive from the vHPC onto mPFC interneurons should reduce the component of mPFC inhibitory synaptic activity that is driven by (and synchronized with) vHPC. In this way, the deficit in vHPC excitation of mPFC interneurons that we found is consistent with the reduction in vHPC-mPFC LFP synchrony we observed. We apologize that this logic was not more explicit in the original manuscript and have added text to expand on this point in the Discussion of the revised manuscript. We do agree that looking at the synchronization between specific prefrontal cell types and the hippocampal theta rhythm is an important future direction and now mention this in the Discussion.

10) Does homecage WPLI predict EPM behavior in individual animals?

At the level of the entire population (WT + Het), there is a non-significant trend towards a predictive relationship, but this is driven entirely by the difference between the WT and Het groups. We have included Author response image 1 showing this.

11) This model is proposed as an alternative to the suggestion that vHPC provides direct excitatory input to pyramidal neurons, but it doesn't address Padilla-Coreano et al., 2016 findings that blocking excitatory input did not affect overall firing rates of either pyramidal or interneurons but decreased firing rates in the preferred arm. By their model, shouldn't firing rates increase in the non-preferred arm?

Our model is that vHPC-mPFC input recruits feedforward inhibition, and this feedforward inhibition suppresses the firing of mPFC neurons, specifically out-of-phase firing that is driven by “noise” input. vHPC-mPFC input also excites pyramidal neurons. Thus, firing rates of excitatory neurons reflect a combination of feedforward excitation and feedforward vHPC-mPFC inhibition, as well as excitation and inhibition from other sources. We hypothesize that when vHPC input is suppressed, total circuit E/I remains relatively constant but that inputs from other sources now become the primary drivers of inhibition. As a result, overall firing rates may not change. However, if the differential firing in the preferred arm relative to the non-preferred arm is driven mainly by vHPC input, then this difference should be suppressed, which is what was observed. In other words, our model emphasizes that feedforward inhibition from the vHPC to mPFC plays an important role in enabling the mPFC to respond appropriately to feedforward excitation in this pathway. However, our model still presumes a key role for vHPC-mPFC feedforward excitation as well.

12) Figure 2: Was the directionality of the HPC-PFC WPLI measure consistent with an effect on HPC drive to PFC?

WPLI is unsigned, i.e., it measures phase locking using the magnitudes of the imaginary component of the phase difference. However, when we examined the signs of these phase differences, we found that when mice were in the open arms, for 5/6 wild-type mice and 6/6 *Pogz* heterozygous mice, the imaginary component was above the x-axis in the complex plane, suggesting that hippocampal activity leads prefrontal activity. We have now included this in the revised text, subsection “*Pogz^+/-^* mice have reduced hippocampal-prefrontal theta synchrony”.

13) Figure 3 needs to be expanded and clarified:a) How was cross-frequency coupling calculated? The Materials and methods state that "Cross-frequency coupling was calculated by comparing the instantaneous phase in a high frequency band with the instantaneous amplitude in a low frequency band". Usually the converse would be true, but still more details are required – was an established method like the Modulation Index used?

The reviewers are correct – there was a typo in terms of the phase/amplitude for high/low frequency activity. We did compute cross-frequency coupling via a standard method and have corrected and expanded our description of this as follows: “Cross-frequency coupling was calculated by comparing the instantaneous phase in a low frequency band with the instantaneous amplitude in a high frequency band. Specifically, instantaneous phase and amplitude were obtained using the Hilbert transform (using the Matlab function *hilbert*). At each point in time, this phase and amplitude were combined to yield a vector in the complex plane. We combined vectors from successive timepoints, and the amplitude of the vector sum was normalized to the sum of all the amplitudes to quantify the strength of cross-frequency coupling.”

b) Although it is possible to understand the results in Figure 3, it is not very reader-friendly. The authors should add some panels explaining each step in this analysis, showing visually how they go from raw LFP recordings to the plots shown in Figure 3.

We have substantially reworked Figure 3 and tried to indicate the workflow with arrows and more thorough descriptions. We believe the new version is more reader-friendly.

c) There is insufficient information given about the empirical salient feature analysis. Figures 3A-D are not labeled in a way that readers can decipher the identities of features. The authors should show how all the features fared throughout the pipeline. Moreover, why doesn't this data analysis strategy pull out the known finding (theta theta phase synchrony)?

First, we have tried to make the features more clear in the revised version of Figure 3. All of the features are listed in Tables 1 and 2. If there is a specific piece of information that the reviewers would like us to provide or plot, we would be happy to do so.

Second, the reviewer raises a good point re: theta phase synchrony. Neither theta phase synchrony, nor theta amplitude covariation is “pulled out” by the ICA. This reflects the fact that over the entirety of the task, theta phase synchrony is being influenced by different factors than this IC, even though theta phase synchrony and this IC both evolve in parallel specifically during closed arm-center zone approaches. In other words, during closed-center runs, both theta phase synchrony and the IC both exhibit a sharp rise followed by a return to baseline. However, during the rest of the task, these two measures must diverge. I.e., closed arm-center approaches recruit both theta phase synchrony and the IC, but there must be other EPM behaviors which differentially recruit these two measures. As we identify more behaviors within the EPM (e.g., behavioral motifs identified via MoSeq, DeepLabCut, etc.), presumably we will identify behaviors for which theta synchrony and this IC diverge. This is an important and ongoing area of work in the lab. We have added text related to all of these issues to the Discussion.

[Editors' note: further revisions were suggested prior to acceptance, as described below.]

Essential revisions:1) Even though it occurred in equal numbers across genotype, a concern was raised that in a paper focused on theta, 38% (8/21) of the mice were excluded for lacking a clear theta peak (despite having good histology). Is there any explanation for why theta is aberrant in such a large fraction of the data? What would happen to the findings if these mice were included? The authors should include any mice that don't have a technical problem with the electrodes or location. An alternative approach to point 1 would be to analyze the 4+4 “non theta” mice separately in a supplement.

We apologize – we obviously were not clear about the details and this led to understandable confusion on the part of the reviewers. We did not exclude mice which lacked a clear theta peak despite their having good histology. Rather we excluded mice once they lacked a clear theta peak and did not include these mice subsequently for histology. The ventral hippocampus is a deep structure, so mistargeting is not uncommon. In tetrode recording experiments electrodes are typically lowered progressively until electrophysiological markers of the pyramidal cell layer are observed. In the case of fixed electrodes this was not an option, so we simply excluded mice which did not have a clear theta peak during periods of locomotion from the subsequent workflow, which included histology.

2) In general, it seems that people tend to treat units recorded from the same animal as independent but consider LFP data recorded with the same electrode as a single biological replicate (and trials as experimental replicates). The authors should use a repeated measures design, use the average LFP per mouse, or some other statistically valid approach for their analyses.

We agree that appropriate statistical comparisons are critical. That being said, we think statistical validity should be determined by mathematical considerations rather than what researchers in the field tend to do. In this context, we are not clear on the specific reason why the reviewers think this comparison was problematic. Specifically, the reviewers suggested that we use a repeated measures design. A repeated measures design accounts for statistical dependencies between groups (caused by one set of variables) in order to compare the effect of a different variable. In this case, measurements from a single mouse are correlated. Therefore, we computed the z-score of each measurement relative to other measurements from the same run (in the same mouse). This removes any statistical dependencies related to the fact that multiple observations come from the same mouse. This seems to be what the reviewers are asking for, and we don’t understand why this would not be statistically valid. We agree that simply taking all the LFP measurements from one group of mice and comparing them to another group of mice would not be valid, but that is not what we did.

We understand what the reviewers are saying – that most LFP studies do not use multiple LFP measurements as separate samples. This is because in those LFP studies there is no way to remove the statistical dependency between different LFP measurements from the same mouse without subtracting off the mean value for that mouse and thus eliminating any between group differences.

Put another way, in most LFP studies, the source of statistical dependencies within each group (mouse by mouse variation) cannot be separated from a variable driving statistical differences between different groups of mice. However, we are not directly comparing an LFP-derived measurement between groups. Rather we are comparing the modulation of this LFP-derived measurement during each run between groups. As such the relevant unit of analysis is a run (not a mouse). Within-run variation is distinct from within-mouse statistical dependencies. Thus, in contrast to most other LFP studies, we are able to remove within-mouse statistical dependencies without affecting between-group differences in our measure of interest.

3) The explanation of Figure 3 is improved, but questions remain about the procedure. It seems that ICs generated from a combination of WT and POGZ mice (should the number of mice be 13?).

Correct – ICs were identified in both WT and Het mice, and the reviewer is correct that the total number of mice should be 13 (we apologize for the typo).

How was consistency across mice measured? By pairwise correlations?

As stated in the text: “To identify similar ICs that were conserved across mice and thus likely to be biologically meaningful, we calculated the correlation coefficient between all pairs of ICs (Figure 3B), then applied a threshold to this pairwise correlation matrix to identify pairs of highly similar ICs (Figure 3C). We then performed clustering on this dataset (Materials and methods) to identify characteristic ICs that appear repeatedly across mice (Figure 3D).”

How did this procedure measure consistency across all mice?

Correlation measures the similarity (i.e., the consistency) between ICs in two different mice. Clusters correspond to groups of ICs which all have high correlation (correlation above a threshold) and thus are all similar (consistent).

Was there any consideration given to whether ICs were consistent within or across genotype?

No. One could have done a different analysis and identified ICs that were consistent within a genotype. However, we did not do this because of the following concern. Suppose there was no meaningful biological difference between two groups of mice. If you identified ICs in one group (e.g., WT mice), then compared the activity of these ICs between the two groups, you might be more prone to find “false positive” differences than if you had derived the ICs using data from both groups. This is because the ICs would be “overfit” to variation within one group. Thus, variation in that group would tend to occur along each IC axis. However, variation in the other group would be less well aligned with each IC axis. In principle, this would lead to behaviorally-driven changes in IC activity being more pronounced in the group used to identify the IC than in the other group, even if these were not really two different groups. Deriving the ICs using mice from both groups should mitigate this potential issue.

4) What percent of high correlation ICs contained a measure of theta synchrony? Similarly, what percent of ICs were task-modulated? Without these measures, it is hard to interpret the significance of a given IC's task modulation and genotype disruption.

We agree that this information is important and should be included. There were three clusters of ICs that were strongly correlated across different mice. The first one, which we originally described, corresponds to cross-frequency phase-amplitude coupling between hippocampal theta and beta or gamma activity in the hippocampus or PFC. Activity of this IC exhibits a marked rise during center approaches which is absent/deficient in Pogz mutant mice. The second one corresponds to cross-frequency phase-amplitude coupling between prefrontal theta and beta or gamma activity in the hippocampus or PFC. Activity of this characteristic IC was not clearly modulated during center approaches and was not significantly altered in Pogz mutant mice. The third cluster corresponds to broadband power across all frequency bands in the hippocampus and PFC. Similar to IC #1, activity of this IC normally increased during center approaches, but this increase was absent/deficient in Pogz mutant mice.

These observations are now presented in Figure 3—figure supplement 1. Taken together, they support our central finding that theta-frequency communication between the hippocampus and downstream structures such as the PFC is behaviorally modulated, and that the normal pattern of modulation is disrupted in *Pogz* mutant mice. Characteristic IC networks are not observed for cross-frequency coupling in other frequency bands (e.g., phase amplitude coupling between alpha-frequency activity and beta or gamma oscillations), and behavioral modulation is observed in the HPC->PFC direction but not in the opposite direction. This shows that our findings are specific for both frequency-band and anatomical pathway. We have added text describing this in the revised manuscript.

5) The explanation of why ICA did not yield theta phase synchrony was unclear. Isn't theta synchrony a stable “network” conserved across mice? Doesn't that serve as an important positive control of this empiric method?

We appreciate that this is complicated, but hopefully this explanation is more clear. An IC is a group of measures that co-vary. That means they co-vary in time within mice. Furthermore, we found that the same set of measures compose an IC across mice. Thus, consistently across mice these features co-vary in time.

Now, suppose that during center approaches/entries, the hippocampal theta rhythm exerts a particularly strong influence on the rest of the circuit. As a result, multiple measures related to this rhythm will evolve in concert during center approaches/entries. I.e., vHPC-mPFC phase synchrony and coupling between the phase of vHPC theta and the amplitude of higher frequency exhibit a similar pattern of increases and decreases, as we observed. However, most of the time, the mouse is doing things other than approaching the center. And during these other periods of time, these measures can diverge, presumably because different factors are driving their respective evolution (e.g., perhaps the amplitude of gamma activity is largely determined by something other than theta phase during these periods). In this manner, vHPC-mPFC theta phase synchrony need not be a part of the same IC as other theta-related measures.

Here is an analogy. Senators Bernie Sanders (pre-2016) and Rand Paul may have occasionally voted in concert because of their shared isolationist and anti-gun control views. If you looked only at bills related to gun control and foreign military intervention, their pattern of voting would appear to be strongly correlated. However, when viewed in totality, their voting patterns would not be well aligned. So too, theta phase synchrony can be aligned with the rest of the IC during center approaches/entries without actually being part of the IC (because IC membership is mainly determined by other periods, during which the mouse is not approaching the center).

[Editors' note: further revisions were suggested prior to acceptance, as described below.]

Essential revisions:1) Reviewers remain concerned about the authors' decision to ignore the recommendation to use a repeated measures analysis for some of the reported results, in which different trials from the same mouse were treated as independent. Based on the statistics currently used, it is not convincing to draw the conclusion that theta band WLPI is task-modulated in WT, but not PogZ^+/-^ mice. If there happened to be a dependency of the trials due to the mouse, then z-scoring wouldn't remove it. In other words, there could be some unknown reason why impairments are strong in some animals and not others (e.g., some interaction between the genotype and individual experiences during development), and z-scoring would not take care of that. Reviewers agree that a repeated measures analysis should be used. Also, for the effects in some frequency bands and not others, again, a repeated measures analysis would be best, as power in different frequency bands within the same recordings is also not independent. It is important to remember that, at eLife, the decision letter containing the consensus review and recommendations is published along with the article. So, we think that readers would wonder why authors did not follow the suggestion to apply a repeated measures analysis, and this could lower confidence in the results and diminish the paper's impact.

We apologize and were not trying to skirt this issue. The statistical test we had done captured some elements of this statistical design (basically comparing one timepoint to the average of all the other timepoints and looking for a genotype effect on this difference), but now we have a better understanding of what exactly the reviewers are looking for and agree that a more standard test would be beneficial. We have used a linear mixed effects model (specifically the fitlme function in Matlab) as specifically suggested in the helpful correspondence. We now include the results of this linear mixed effects model statistical analysis (using mouse, genotype, timepoint, and genotype X timepoint as fixed factors and run as a random factor) in the legend for Figure 2B and Table 3. Specifically, the p-value for the genotype X timepoint interaction was 0.00039.

We have done a similar analysis for the IC projections and report these results in the legends for Figures 3E (IC #1) and 3-1F (IC #3). For IC #1 the p-value for the interaction term is 0.01. In the case of IC #3, the p-value for the interaction term is 0.052, which is just above the threshold for significance – we do not make any claims about whether or not IC #3 differs across genotypes and have simply provided this information as previously requested by the reviewers, to provide context for our finding re: IC #1.

Re: the WPLI when mice are the home cage – in this case, there is a significant difference between home cage theta band WPLI for WT vs. Het mice. However, when we did an ANOVA, we did not find a significant genotype X frequency band difference. We could remove this data. However, we specifically focused on theta-band synchrony because this has previously been implicated in avoidance behavior in the EPM in multiple studies from multiple labs. Given that we found that theta band WPLI is deficient in Het mice during approaches to the center of the EPM, and that theta band WPLI is also deficient in Het mice in the EPM overall (i.e., irrespective of whether mice are in the open or closed arms), it would be natural for readers to be interested in whether there is also a deficit in theta band WPLI in the home cage. Therefore, we have re-arranged this figure (Figure 2) and now simply present the home cage theta band WPLI, after presenting the data about theta band WPLI in the EPM.

2) The authors should also include the exact statistics and degrees of freedom, APA style, not just p-values when reporting effects. Also, reviewers think it would be best to request that all the figure legends list N, name of statistical test, and p value, so that it is easier for readers to understand how data were analyzed. Currently some figures (such as Figure 1) list only p values.

We have added information on test name, exact statistic, Ns, and degrees of freedom to the figure legends (for Wilcoxon rank sum tests, we simply report the N’s as there is no specific degrees of freedom apart from the N).

3) The final N for Figure 3E and Figure 3—figure supplement 1D-F remains unclear. Are these N also based on z-scored trial data? If so, they are subject to the same statistical concern mentioned above.

These N’s are based on runs – we have made this more clear in the revised figure legends and Table 3, and have performed a linear mixed effects model analysis, as described above in our response to point 1.